# Spatiotemporal functional organization of excitatory synaptic inputs onto macaque V1 neurons

Niansheng Ju[1,2,3,4], Yang Li[1,2,3,4], Fang Liu[1,2,3,4], Hongfei Jiang[1,2,3,4], Stephen L. Macknik [5], Susana Martinez-Conde[5] & Shiming Tang[1,2,3,4]*

The integration of synaptic inputs onto dendrites provides the basis for neuronal computation. Whereas recent studies have begun to outline the spatial organization of synaptic inputs on individual neurons, the underlying principles related to the specific neural functions are not well understood. Here we perform two-photon dendritic imaging with a genetically-encoded glutamate sensor in awake monkeys, and map the excitatory synaptic inputs on dendrites of individual V1 superficial layer neurons with high spatial and temporal resolution. We find a functional integration and trade-off between orientation-selective and color-selective inputs in basal dendrites of individual V1 neurons. Synaptic inputs on dendrites are spatially clustered by stimulus feature, but functionally scattered in multidimensional feature space, providing a potential substrate of local feature integration on dendritic branches. Furthermore, apical dendrite inputs have larger receptive fields and longer response latencies than basal dendrite inputs, suggesting a dominant role for apical dendrites in integrating feedback in visual information processing.

[1] Peking University School of Life Sciences, 100871 Beijing, China. [2] Peking-Tsinghua Center for Life Sciences, 100871 Beijing, China. [3] IDG/McGovern Institute for Brain Research at Peking University, 100871 Beijing, China. [4] Key Laboratory of Machine Perception (Ministry of Education), Peking University, 100871 Beijing, China. [5] State University of New York, Downstate Health Sciences University, 11203 Brooklyn, NY, USA. *email: tangshm@pku.edu.cn

Cortical neurons sample from their dendritic synaptic inputs as the basic unit of computation[1]. Whereas in vitro patch-clamp and intracellular recordings have advanced our understanding of how dendritic synaptic integration occurs in single neurons[2,3], and in vivo studies have extended those methods to highlight how dendritic activity contributes to cortical functions such as orientation and direction selectivity[4,5], sample size and spatial resolution have limited the utility of these methods. High-resolution two-photon calcium imaging of dendrites, in contrast, can achieve long-term functional mapping of individual dendritic inputs in the intact brain[6–13]. As the temporal sequence of synaptic inputs has been verified to be of vital importance to dendritic nonlinear integration[14,15], high-speed dendritic imaging can provide important new insights into dendritic computational mechanisms. A recently developed genetically encoded glutamate-sensing fluorescent reporter, iGluSnFR[16], which has a high signal-to-noise ratio (SNR) and fast kinetics, promises to map the spatiotemporal functional organization of dendritic excitatory inputs. In this study, we performed two-photon dendritic imaging with iGluSnFR in awake macaque monkeys, and obtained fine functional spatiotemporal maps of dendritic excitatory inputs in individual V1 neurons.

## Results

**Measuring excitatory synaptic inputs onto dendrites of macaque V1 neurons.** We performed two-photon imaging on dendrites of sparsely labeled individual V1 neurons expressing SF-iGluSnFR.A184S (Fig. 1a and Supplementary Fig. 1) while monkeys fixated and viewed visual stimuli. High-resolution dendritic imaging was conducted using a 25× objective lens (1.1 N.A., Olympus) in 4× zoom-scanning mode, covering a field of view of 136 × 136 μm. The entire set of 81 visual stimuli consisted of either 1 of 9 different color patches, or of a drifting grating with 1 of 12 orientations, 1 of 2 drift directions, and 1 of 3 spatial frequencies (Fig. 1c, bottom). Robust and spatially localized fluorescence increases were evoked on dendrites (Supplementary Fig. 2 and Movies 1 and 2), which we refer to as regions of interests (ROIs) (see Methods for analysis details). The background fluorescence changes were minute compared to the robust signals we found on our dendritic recordings (Supplementary Fig. 2c–g). We note that a few hot-spots outside of the targeted dendritic regions exhibited strong activity during visual stimulation, though these untargeted regions could be reliably excluded from our analysis in each single neuron (see Methods for details of our region-of-interest (ROI) analysis). Individual inputs showed significant tuning to orientation, spatial frequency or color (Fig. 1c, d). Given that the stimuli were presented randomly, and taking into account the high repeatability of the single-trial fluorescence traces (Supplementary Fig. 2h–k), our data indicate that the input signals we recorded did not arise from brain motion related artifacts.

Spatial resolution is important for high-quality sampling of spine data, and here we show that our imaging resolution is sufficiently high to tease apart different ROIs. Our two-photon microscope was calibrated to capture high-resolution two-photon images: The point-spread function (PSF) was measured at 0.4 μm X-Y plane resolution, and 2.0 μm z-axis, when using the 25x Olympus objective used for these experiments. This is a typical range of measurements compared to other work in the field, and we have achieved the same diffraction-limited microscopic resolution—near the physical limit also found in other similar two-photon systems—reported in prior imaging studies of dendrites, spines, and other filaments and small cells in the neuropil. In our dendritic imaging experiments, the sampling resolution was set to about 4 pixels per μm. This setting magnified

beyond the physical limitations of the two-photon imaging system. This standard approach[10–12] provided single-ROI signals that were robust and repeatable (Fig. 1 and Supplementary Fig. 2). We also replicated these results using a higher power objective combined with higher magnification to achieve higher spatial resolution (18 pixels per μm), using a 60× objective lens under zoom 8× and 512 × 512 pixels. As those recordings produced similar data, we may conclude that the results achieved with the 25 × objective have the necessary precision for spine analysis, and thus represent accurate and valid findings.

To estimate the contribution of z-axis fluorescence leaks across different depths, we captured a series of continuous Z-stack images having 3 μm steps along the z-axis (Supplementary Fig. 1e). Each step varied greatly and thus there was no mixing beyond the z-axis diffraction-limited resolution of our 25× objective (2 μm), establishing conclusively that the fine imaging resolution along the z-axis was <3 μm. Notably, to maximize our yield of dendritic inputs in each experimental session, we targeted focal planes with the longest possible continuous dendritic trees (as an example, we set the imaging depth at 127 μm for the neuron in Supplementary Fig. 1e). Hence, the high continuity of the dendritic tree arises from our selection of the depth plane to image from, and from the intrinsic neuronal morphology of pyramidal dendrites, rather than from a wide depth-of-field or from Z motion.

We found that even adjacent ROIs could possess a wide variety of preferences between and within domains of orientation, spatial frequency (SF) and color (Figs. 1c, d and 2b–d). That would not be the case if there was significant spillover from nearby synapses (either on the neuron we recorded from or on nearby neurons). Responses of individual ROIs exhibited a high-degree of stability across trials (mean trial-to-trial correlation value to tuned stimuli $\pm$STD = 0.70 $\pm$ 0.14, $n = 21$ trials, whereas untuned stimuli exhibited negligible correlation = 0.04 $\pm$ 0.30; Fig. 1e, f and Supplementary Fig. 2i–k). Across the 1818 dendritic ROIs obtained from three monkeys (from 23 neurons, with 79 $\pm$ 29 dendritic ROIs per neuron, mean $\pm$ STD), the mean ROI response intensity was 0.36 $\pm$ 0.03, 0.41 $\pm$ 0.04, and 0.31 $\pm$ 0.05 for each monkey ($\Delta F/F_0$, mean $\pm$ STD, $n = 6,8,9$ neurons respectively; Fig. 1g). To validate the inputs as excitatory glutamatergic (Glu) signals, we co-expressed RCaMP in the same neurons, and collected combined Ca-imaging and Glu-imaging signals from both indicators (Supplementary Fig. 3). Consistent with a previous dendritic study[10], our data showed that summed input orientation preferences matched the orientation preference of the integrate somal Ca-response, and that summed input orientation selectivity was moderately broader than somatic orientation selectivity. Thus, high-quality dendritic imaging of glutamate responses in awake macaque monkeys enabled precise determination of the functional properties of excitatory synaptic inputs.

**Synaptic inputs are spatially clustered by individual visual features.** We first examined how excitatory synaptic inputs—tuned to each visual property tested—are spatially arranged on dendritic shafts. We found moderate clustering within local dendrites along single feature dimensions of receptive field (RF) and orientation ($\lambda$ = 4.9 and 3.3 μm respectively, see Methods for analysis details; Fig. 2f), as well as SF and color ($\lambda$ = 4.3 and 5.6 μm). Recent two-photon dendritic imaging studies have revealed that, despite understandable excitement in the field concerning the discovery of functional input clustering[11,12,17,18], highly scattered distributions of inputs are also common[6–8,11,13]. There is no known explanation for this difference, but combined with our results in macaques, the evidence suggests that dendrites on visual neurons within orientation maps receive functionally clustered synaptic inputs[12], whereas neurons that are not

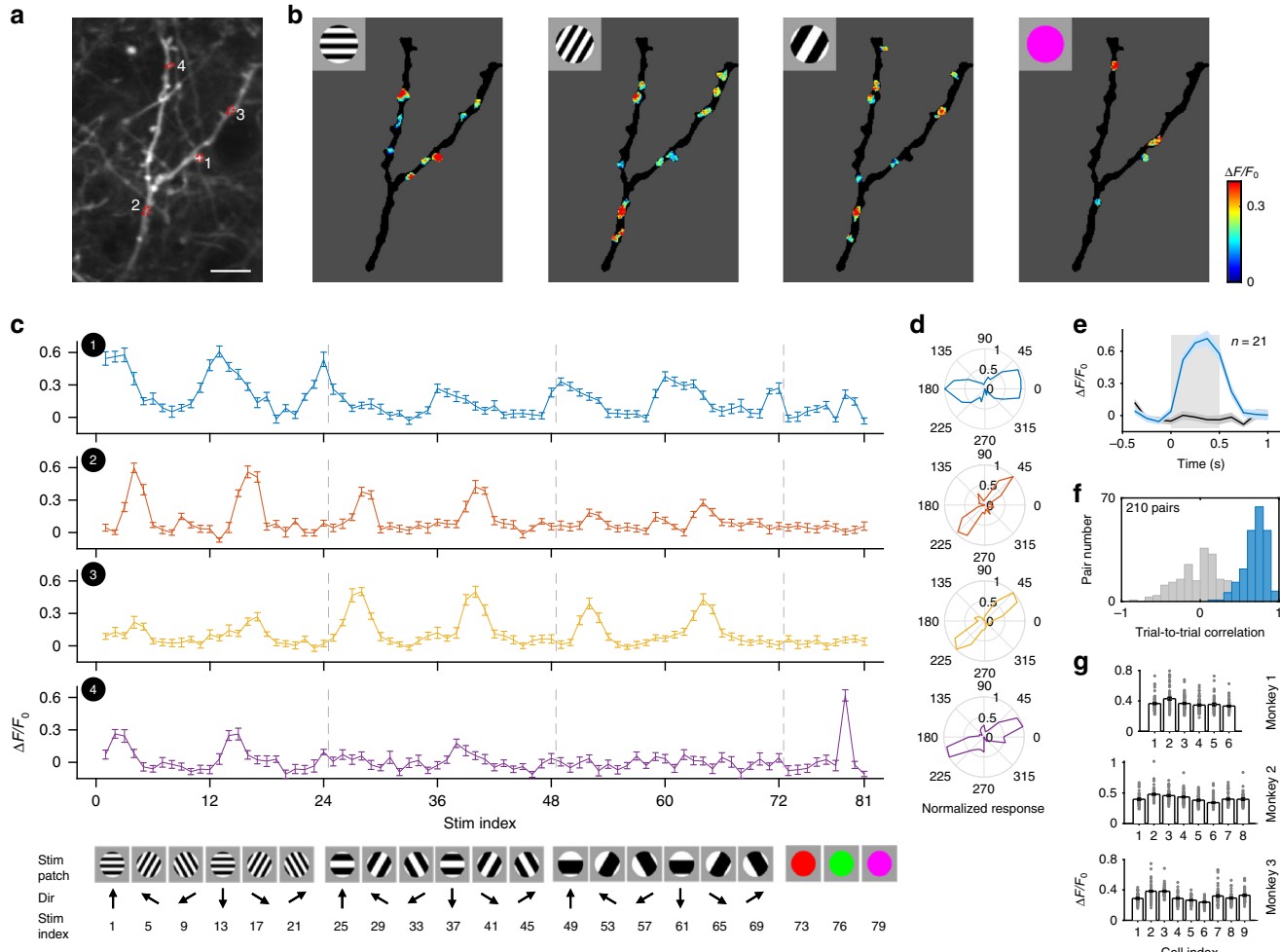

**Fig. 1 Imaging excitatory inputs on dendritic shafts in awake macaque monkey V1. a** Dendritic shafts of one neuron expressing SF-iGluSnFR.A184S in superficial layer of V1, recorded at a depth of 141 μm on Day 84 post-injection. The image was obtained by averaging 1000 frames from one recording session. Red dotted circles are representative ROIs; scale bar, 10 μm. **b** Fluorescent responses (averaged across 21 repeats) on dendritic shafts elicited by visual stimuli (insets). Only robust input activities within regions (of >2.5 STD over baseline) were analyzed. Intensity of iGluSnFR fluorescence signals are denoted by the colorbar (right). **c** Responses ($\Delta F/F_0$ averaged across 21 repeats) from the numbered ROIs from panel **a**, plotted as a function of stimulus (below; error bar, SEM). Top, response curves; bottom, a subset from the entire set of 81 visual stimuli. **d** Orientation polar plots from the optimal SF for each ROI. **e** Fluorescence traces of ROI 1 corresponding to the highest (Stim index = 13, blue) and lowest (Stim index = 81, black) response intensity from all stimuli tested. Curve shadow, SEM; square shadow, stimuli onset interval. **f** Repeatability of worst (gray) and best (blue) stimulus responses across trials as in **e** were calculated as trial-to-trial correlations. **g** Mean value of the optimal response to each of the 1818 ROIs averaged with each the 23 source neurons (presented separately by monkey; error bar, SEM).

embedded within orientation mapped cortex have less clustering in their synaptic inputs[6,8,11].

To investigate the relationship between the functional properties of synaptic inputs into neurons as a function of their relative position within orientation maps, we obtained maps of orientation pinwheel structures using a large field of view: a 16× objective lens (0.8-N.A., Nikon) with no optical zoom (1×) that had a field of view of 850 × 850 μm (Supplementary Fig. 4). We found that the summed orientation preference of dendritic inputs to individual neurons tended to match the local orientation columnar maps ($p < 10^{-9}$, linear regression; Fig. 3a and Supplementary Fig. 5a). Also, input preferences varied in consistency as a function of the cortical column orientation preference map gradient ($p = 0.05$, linear regression; Supplementary Fig. 5b). In summary, synaptic inputs onto neuronal dendrites tend to exhibit the functional properties consistent with local orientation columnar maps, including orientation preference and its diversity.

**Functional integration and trade-off between orientation/color inputs in basal dendrites of individual V1 neurons.** Whereas most neurons located in V1 layer 4 have been classified as either orientation- or color-selective, a large percentage of upper-layer V1 neurons exhibit chromatic-orientation feature integration[19–21]. Consistent with previous studies, we found that nearly all individual neurons received both abundant orientation- and color-selective inputs on their basal dendrites (72% vs. 57%, 93% vs. 37% and 80% vs. 34% from the three monkeys, respectively; Figs. 2e and 3b). Orientation-dominant and color-dominant inputs were interdigitated on individual dendrites (area under curve (AUC) of ROC = 0.45 ± 0.09, mean ± STD across 23 neurons from three monkeys; Supplementary Fig. 6). The orientation vs. color index, a measure of each ROI's response to orientation vs. color stimuli (more details in Methods) is shown here. The proportion of orientation/color mixing varied across individual neurons, and correlated with the position of the neurons' somas within the cortical orientation map (Supplementary Fig. 5c). Neurons in

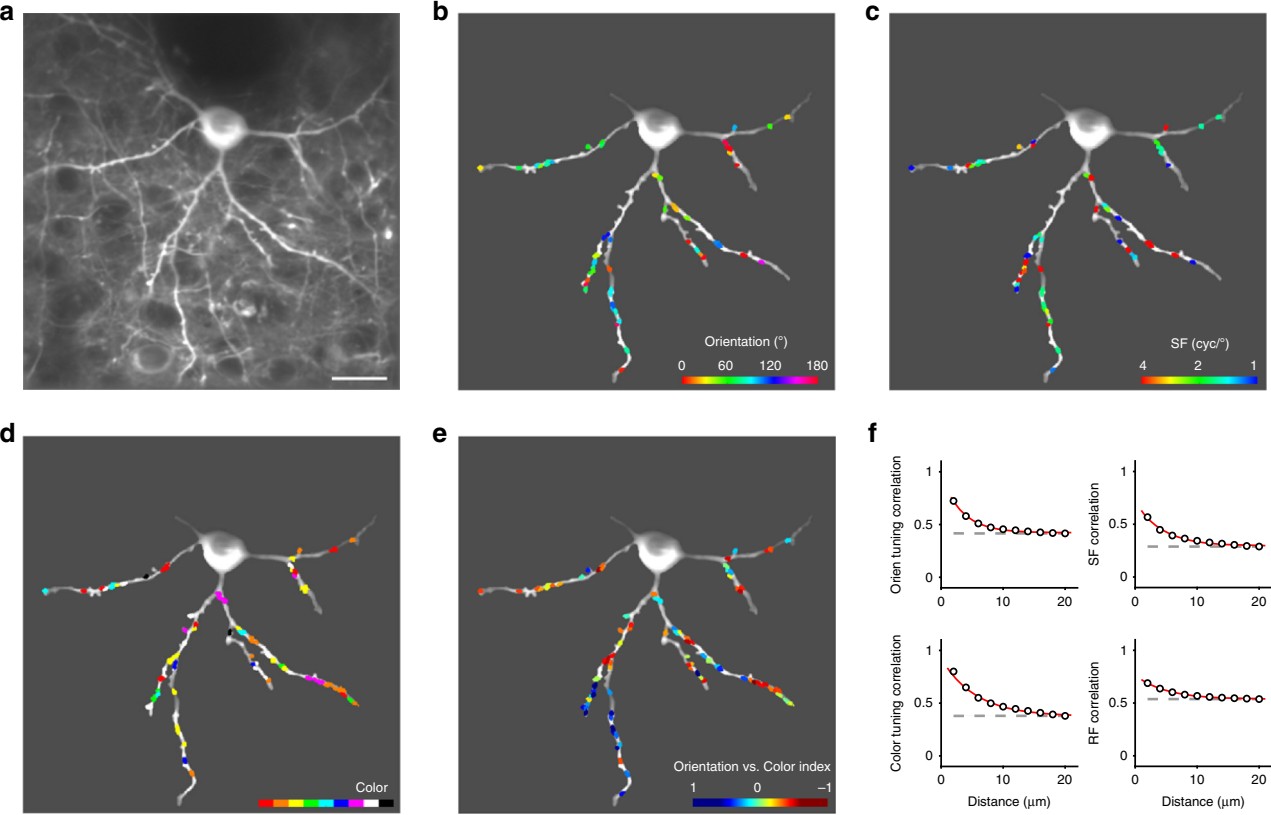

**Fig. 2 The spatial organization of dendritic excitatory inputs on individual V1 neurons. a** Two-photon image of an example neuron (depth of 127 μm on Day 71 post-injection) expressing SF-iGluSnFR.A184S in V1 superficial layers. The image was averaged across 1000 frames. Scale bar, 20 μm. **b** Map of orientation-selective ROIs on dendrites of the example neuron. Orientation preferences are colored for vector sum polarity for each ROI (refer to colorbar at the lower right). **c** Map of SF preferences of the orientation-selective inputs. SF preferences are colored for fitting maximum for each ROI. **d** Map of color-selective inputs. Color preferences are labeled with the dominant polarity for each ROI. **e** A functional preference map of dendritic inputs. ROIs are colored for the orientation vs. color index. **f** Relationship between the dendritic distance of ROI pairs (18,581 ROI pairs in total from 23 neurons) and their tuning correlation coefficients (top left, orientation tuning correlation among orientation-selective ROIs; top right, SF tuning correlation among orientation-selective ROIs; bottom left, color tuning correlation among color-selective ROIs; bottom right, RF correlation among orientation-selective ROIs). Data points are fitted with an exponential curve (red), and shuffled data are presented as a reference (gray dashed). Error bar, SEM.

the orientation column iso-domain (with a low local gradient of orientation preference) tended to receive more oriented inputs, and neurons in blobs tended to receive more colored inputs.

Further, input homogeneity correlated with the proportion of each input type for each neuron. Neurons with relatively large amounts of orientation-selective inputs tended to receive homogenous orientation-selective synaptic inputs ($p = 1 \times 10^{-4}$, linear regression; Fig. 3c), and vice versa for color-selective inputs ($p < 0.01$, linear regression; Supplementary Fig. 5d). This orientation/color trade-off suggests a potential functional competition between oriented and colored inputs within the dendrites of individual V1 neurons. It is possible that these measures are underestimates, since we used visual stimuli that were either monochrome gratings or full-field colors in our experiments, rather than colored gratings, which could have more precisely covariance of color and orientation. The present findings concerning the color/orientation trade-off on dendrites may guide future research focused on color/orientation co-tuning. We note, however, that because orientation-color co-tuned inputs are by definition both orientation-selective and color-selective, our conclusion about the integration in dendritic inputs remains valid. Further, previous studies found color-orientation feature integration only in Layer 2/3 neurons[19–21], and a recent study using cellular-resolution two-photon imaging further revealed that orientation and color are jointly coded with individual

V1 superficial neurons[22]. Our results thus corroborate these studies and moreover present direct evidence of dendritic level computations for orientation and color integration mechanisms.

**Synaptic inputs are functionally scattered in multidimensional feature space.** Synaptic inputs had a propensity to spatially cluster on dendrites with respect to individual visual features, while tending to be tuned across an array of feature dimensions. Further, each input had a unique combination of specific feature preferences (including orientation, SF and RF; Figs. 1c, 3a, b), dispersed across the multidimensional feature space. Thus, one local pair of synaptic inputs might share similar orientation preferences while possessing quite dissimilar RFs (Fig. 4a). In contrast, another pair of synaptic inputs might have both different orientations and dissimilar RFs (Fig. 4b), or different orientations and similar RFs. Thus, pairwise correlations for each input pair's preferences for orientation, SF and RF, revealed substantial scatter (mean $R^2 = 0.03$, $R^2 = 0.04$, and $R^2 = 0.01$ for SF vs. orientation, RF vs. orientation and RF vs. SF, respectively; Fig. 4c–e). In some cases, synaptic inputs onto an individual neuron were quite homogenous within a feature dimension, but heterogeneous in other feature dimensions. This wide scattering served to maximize the pool of potential matches between dissimilar features within local dendritic branches (see Fig. 4a, b). As such, it provides a potential

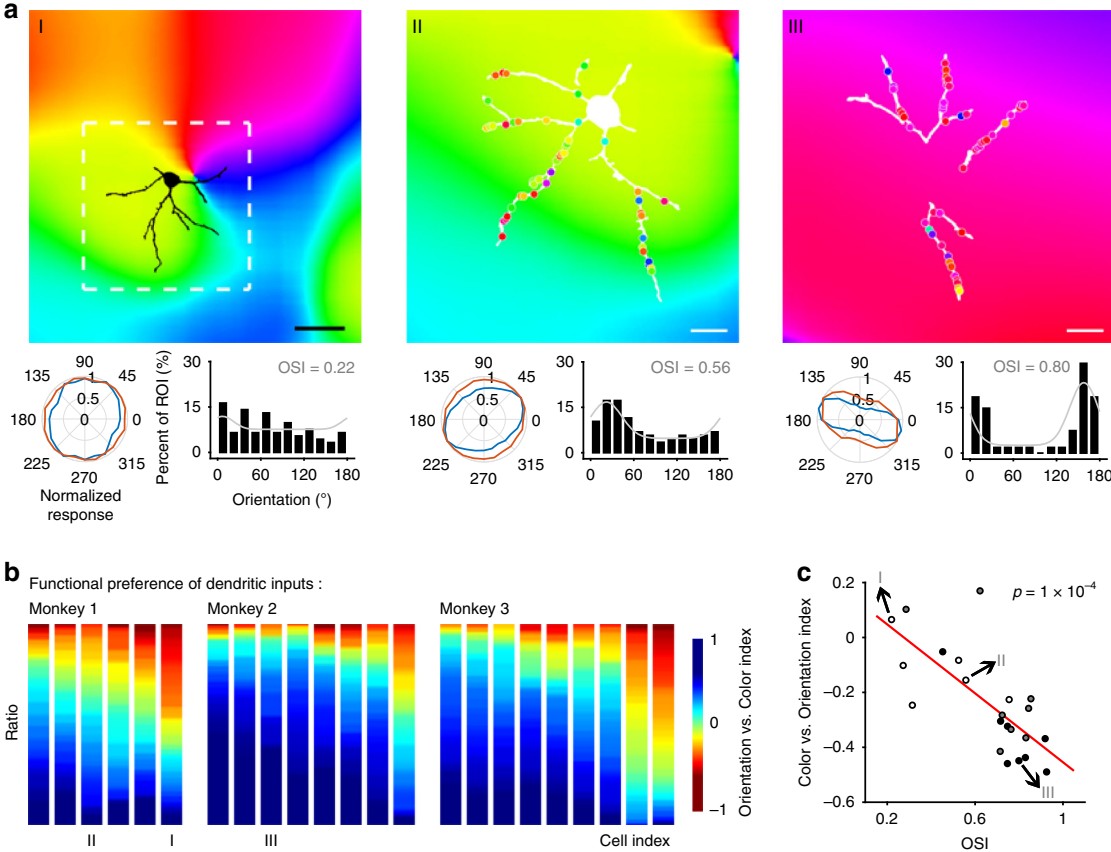

**Fig. 3 Functional integration and trade-off between orientation/color inputs in individual V1 neurons' basal dendrites. a** Left panel, top, neuron position on local orientation map (measured with global iGluSnFR fluorescence, depth of 127 μm on Day 71 post-injection). White dashed square, 166 μm in width and height, corresponding to 100 pixels in raw two-photon images; scale bar, 50 μm. Left panel, bottom left, normalized polar plots of orientation tuning of summed ROI responses (blue lines) and local cortical orientation selectivity (averaged response within a local area marked by the white dashed square; red lines); bottom right, frequency distribution of ROIs' preferred orientation fitted with Gaussian functions (gray curves). Orientation selectivity index (OSI) of the fitted curve represented input consistency. Right two panels, dendritic orientation map of another two sample neurons (recorded at depths of 206 and 286 μm on Day 74 and Day 91, respectively) overlaid on local population orientation maps. Scale bar, 20 μm. **b** Statistics of functional preference between orientation vs. color for all 23 measured neurons, sequenced by the average orientation vs. color index value. **c** Color vs. orientation index for each neuron vs. its OSI.

computational substrate for multidimensional feature integration at the dendritic level in V1 superficial layer neurons.

**Distinct functional organization of apical and basal dendritic inputs**. Excitatory synaptic inputs onto apical dendritic tufts, which contain a mass of convergent long-range cortical and subcortical projections[23,24], are thought to be critical to receiving feedback from higher level brain regions[25,26]. Recent in vivo experiments combining two-photon calcium imaging and intrinsic optical imaging have functionally compared V1 neuronal responses to feedback[27,28]. However, no studies to date have performed a direct comparison of functional inputs to apical dendrites vs. those to basal dendrites in single neurons. We compared apical vs. basal inputs in individual neurons with respect to their functional preferences and response latencies (Fig. 5a). Apical and basal dendrites received a similar proportion of orientation-selective vs. color-selective inputs ($81 \pm 18\%$ oriented inputs on AD, $87 \pm 13\%$ on BD, mean ± STD, $n = 13$ neurons; $p = 0.37$, Wilcoxon Rank-Sum Test), and had similar summed orientation preferences ($R^2 = 0.87$ fit to the unity line; Fig. 5b, c). Consistent with previous observations of larger RFs in V1 layer 1 feedback boutons[27], we found that apical inputs had significantly larger mean RF sizes than basal inputs ($0.70 \pm 0.16$ square degree on AD vs. $0.31 \pm 0.13$ on BD,

mean ± STD, $n = 9$ neurons; $p = 9 \times 10^{-7}$, paired $t$-test; Fig. 5b, d). We acquired precise latencies for apical vs. basal inputs by imaging small regions at 226 Hz ($68 \times 68$ μm with $64 \times 64$ pixel resolution), and found that apical input latencies were on average 10 ms higher than basal input latencies ($92 \pm 12$ ms for AD and $82 \pm 8$ ms for BD, mean ± STD, $n = 13$ neurons; $p = 0.008$, paired $t$-test; Fig. 5e). The combination of larger RF sizes and higher response latencies indicate that feedback dominates apical inputs.

**Discussion**

Our findings provide a novel approach to identify the spatiotemporal organizational principles for excitatory dendritic inputs to V1 superficial neurons of awake macaque monkeys. Whereas dendritic imaging techniques were previously applied in anesthetized lower mammals with calcium indicators[5–13], no prior research had ruled out the potentially significant effects of anesthesia on spontaneous neuronal firing rates, or the level of inhibition on dendritic membrane properties[1], while isolating purely excitatory inputs onto dendrites. Critically, no mammals other than old world primates have homologous visual processing capabilities to those of human beings, with respect to color, acuity, and visual cognition[29]. Our use of the glutamate sensor

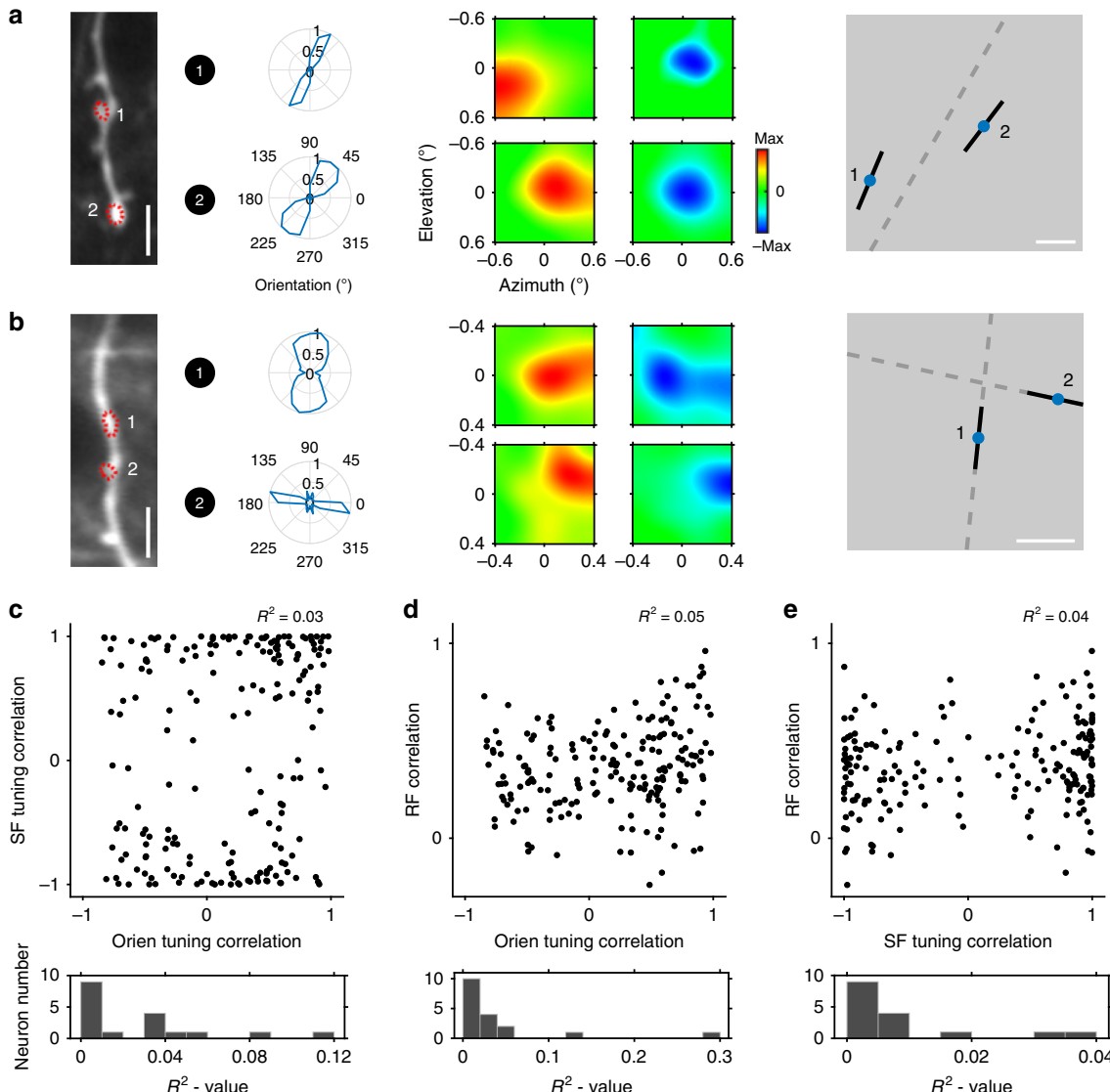

**Fig. 4 Diversity of dendritic inputs among multi-feature dimensions. a** Two ROIs on one dendritic draft share similar orientation preferences while having distinct RFs. Left, sample dendrite (averaged across 1000 frames at 114 μm on Day 86) and target ROIs. Scale bar, 5 μm. Middle, the corresponding orientation preferences and RFs (red, ON; blue, OFF). Right, preferred orientation (black) overlaid at the site of each RF center (blue) for each ROI. Scale bar, 0.2 degree. **b** Another sample dendrite (averaged across 1000 frames at 127 μm on Day 71) with an associated ROI pair. Those ROIs have both different orientation preferences and different RFs. **c** Inter-ROI SF tuning correlation as a function of orientation tuning correlation. Top, scatter plot from a sample neuron; $R^2$ value from linear regression. Bottom, distribution of $R^2$ value from all collected neurons. **d** Inter-ROI RF correlation vs. orientation tuning correlation. **e** Inter-ROI RF correlation vs. SF tuning correlation.

iGluSnFR removed a technical hurdle that was extant in previous studies, by eliminating the need to post-process our data to control for the effects of back-propagating spikes, which are a source of noise in Ca-imaging experiments. In our hands, iGluSnFR resulted in strong signals on neuronal dendrites, whereas dendritic Ca-imaging in dendrites largely failed to provide strong signals in macaque monkeys (Supplementary Fig. 7). Somal imaging of GCaMP6s Ca-indicators nevertheless succeeds in macaque V1[30]. One possible explanation for this discrepancy is that macaques have relatively reduced ion channel densities in V1 pyramidal dendrites;[31] thus, high-quality afferent Ca-signals will be difficult to record with a high signal-to-noise ratio. This interspecies difference highlights the potentially different fundamental computational functions of dendrites in primates vs. those in lower mammals.

iGluSnFR has a high affinity to glutamate, which could potentially make it even sensitive to glutamate arising from spillover between nearby synapses. Note that whereas the $1/e^2$ decay distance for iGluSnFR was found to be as much as 3.6 μm, as illustrated by Kazemipour et al.[32] (corresponding cross-validation tuning correlation value higher than 0.06), the half-life (1/2) decay distance is <1 μm (correlation value higher than 0.22). In other words, whereas iGluSnFR can be used to detect glutamate spreading from up to 3.6 μm away, robust detection of glutamate occurred primarily within 1 μm. In our experiments, we did not detect significant spillover effects between nearby synapses as measured with differential stimulation. Further, significant spillover would result in homogenous nearby synaptic responses, whereas we found great diversity between even neighboring synapses stimulated at high-strength. Thus, our results are consistent with Kazemipour et al.'s findings, and we obtained similar sizes of inputs onto neuronal dendrites from our iGluSnFR imaging experiments (the mean value of ROI radius equaled 0.76 μm).

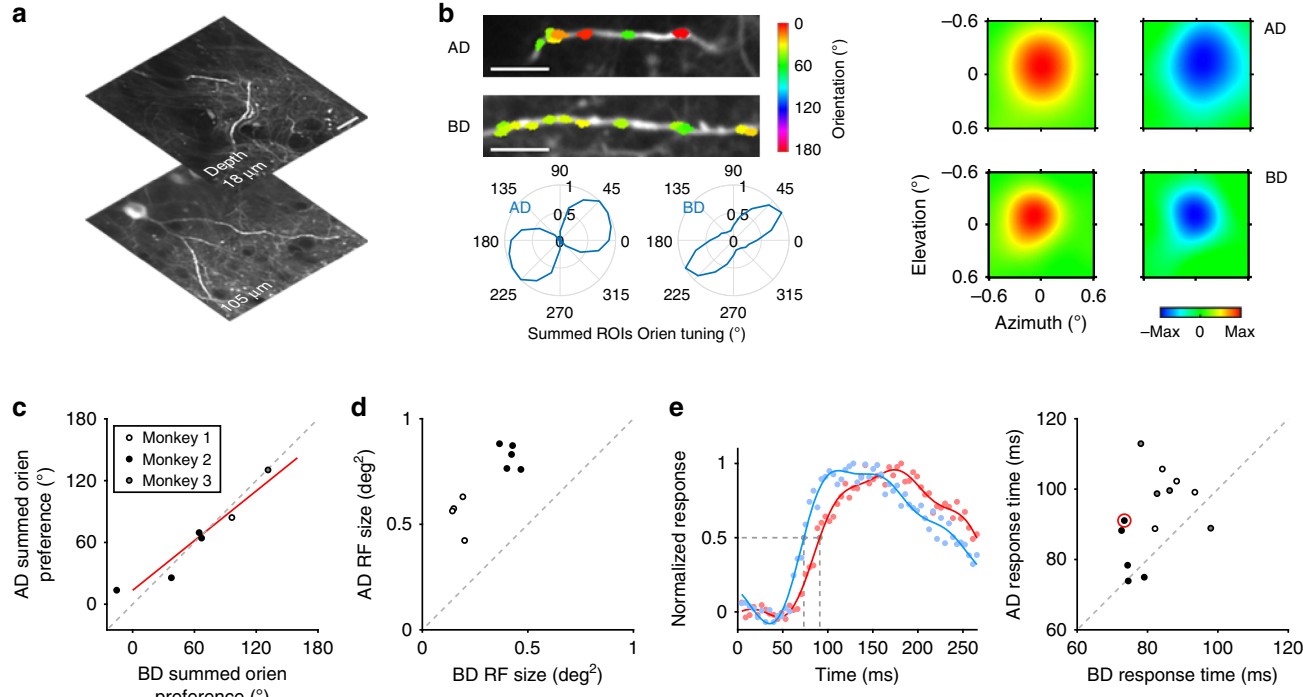

**Fig. 5 Comparison of excitatory dendritic inputs on apical dendrites vs. those on basal dendrites. a** Apical and basal dendritic shafts of one example neuron (recorded at depths of 18 and 105 μm, respectively, on Day 88). The images were averaged across 1000 frames. Scale bar, 20 μm. **b** Orientation preference and RF of dendritic inputs on AD and BD. Top left, orientation preferences of ROIs on AD and BD, respectively, colored for vector sum polarity for each ROI. Lower left, summed orientation preference of all orientation-selective ROIs on AD and BD respectively (AD, 18 ROIs on two dendrites; BD, 43 ROIs on three dendrites). Right, example RFs of two ROIs on AD and BD, respectively. **c** Summed orientation preference of orientation-selective ROIs on AD and BD of each neuron (neurons with <15 ROIs on their AD were discarded). Red line, linear fit ($p = 2 \times 10^{-3}$); gray dashed line, unity line. $R^2 = 0.87$ when fitted to the unity line. **d** AD RF size (mean of ON and OFF RF sizes) vs. BD's of each neuron. **e** Time courses of response on AD vs. BD. Left, normalized fluorescence traces on AD (red) and BD (blue) of the example neuron. Dots, raw data points; lines, Fourier fit; gray dashed lines, marking response time points according to half-height. Right, AD response time vs. BD's of each neuron. The example neuron was designated with a red circle.

In this study, we obtained high-quality sparse labeling of iGluSnFR sensor with hSyn promoter in macaque visual cortex (Supplementary Fig. 1a). We also tested traditional sparse labeling strategies (both Cre and tTA with low concentration), but found that hSyn achieved the highest-quality sparse labeling. We assume this is due to low infection efficiency, in macaque cortex, of the specific virus we employed (AAV1.hSynap.SF-iGluSnFR.A184S. WPRE.SV40, derived from the Looger lab and Addgene). Other species-based differences between murine and primate genetic labeling methods have been found previously[33]. The depth of our collected neurons ranged from 100 to nearly 400 μm, and we recorded neurons with both spiny and smooth dendrites (Supplementary Fig. 1b). By employing the hSyn promoter, which is not specific for cell type, we could not determine specific cell types as a function of our labeling, though we could confirm that we recorded from visually responsive neurons. It remains a major technical challenge in the field to target specific types of neurons within macaque cortex in vivo[34,35]. Owing to the technical limitations of genetic selection within macaque cortical neurons, nearly all two-photon imaging studies in this field can make only general conclusions across different types of neurons[22,33,36,37], and this remains true for our work.

Consistent with previous results that superficial V1 neurons encode both orientation and chromatic selectivity, we found functional integration and trade-off between orientation and color inputs in V1 neuron dendrites. Critically, the higher the proportion of orientation-selective inputs received by a given neuron, the more uniform those inputs tended to be, which presumably contributed to the provenance of orientation selectivity in V1 neurons[9,10]. The data presented in the main figures

came from saturated color stimuli. At the beginning of our experiment (on Monkey 1 and Monkey 2), we only used saturated colors. We later realized the need to eliminate the effect of luminance in our experiment, and conducted control experiments to compare the difference between saturated vs. equiluminant color-evoked dendritic responses (on Monkey 3). Color preferences of 79% dendritic inputs stayed stable when comparing saturated vs. equiluminant stimuli (see Supplementary Fig. 8, bottom panel, for error tolerance between neighboring colors, red shadow). Moreover, the averaged response intensity of dendritic inputs did not vary significantly under saturated vs. equiluminant chromatic stimuli ($p > 0.05$ by Wilcoxon Rank-Sum Test). This result makes intuitive sense, given that our saturated condition was not much brighter than the equiluminant condition (as the luminance of each equiluminant color was normalized to the brightest available luminance in the blue channel: the weakest color channel in our monitor, see Methods part for more details). Thus, we did not expect a significant difference between the saturated and equiluminant condition responses.

We found that dendritic inputs had a wide scattering of functional properties within the multidimensional feature space of the visual system. Individual V1 neurons receive thousands of synaptic inputs, which seems redundant if they all shared similar feature properties. The organization patterns revealed in our data may facilitate integration both within and across visual feature domains, and provide a dendritic computational mechanism for neuronal multidimensional feature integration.

Our finding that the latency of apical inputs is temporally lagged compared to basal inputs further indicates that apical dendrites receive feedback inputs, whereas basal dendrites receive

feedforward inputs within the visual hierarchy[27,28], and brings additional insight into the function of the most iconic feature of cortical pyramidal cells. It remains a challenge to directly dissociate feedback vs. feedforward inputs, and our results provide indirect evidence for the provenance of the signals. Future studies will aim to obtain direct evidence by combining new techniques, such as optogenetics or electrophysiology, and specific behavioral tasks, such as a top–down modulated selective attention task. Our functional observations, in combination with the first successful application of a newly developed glutamate sensor in behaving non-human primates, provide a bridge to deeper understanding of the dendritic mechanisms and computational principles of visual information processing.

## Methods

**Monkey preparation.** All experimental protocols followed the Guide of Institutional Animal Care and Use Committee (IACUC) of Peking University Laboratory Animal Center, and were approved by the Peking University Animal Care and Use Committee (LSC-TangSM-5). Rhesus monkeys (*Macaca mulatta*) were purchased from Beijing Prima Biotech, Inc. and housed at Peking University Laboratory Animal Center. Three healthy adult male monkeys (aged 3–4 years and weighing 4–6 kg) were used in this study.

We performed two sequential sterile surgeries for each animal under general anesthesia. In the first surgery, a 20 mm craniotomy was created on the skull over V1 and the dura was then opened. A round cover glass (6 mm in diameter) with a small pore (0.3 mm) was used for targeting the injection pipette and stabilizing the cortical surface during the injection procedure. The injection pipette was made from a quartz pipette (QF100-70-7.5, Sutter Instrument, USA) and pulled with a 15–20 μm tip using a laser-based pipette puller (P-2000, Sutter Instrument, USA). Through the pipette, 120 to 150 nl of AAV1.hSynap.SF-iGluSnFR.A184S.WPRE. SV40 (HHMI Janelia Research Campus and Addgene, titer 2.7e13 GC per ml) were pressure-injected at a depth of ~350 μm. After AAV injections, we inserted a GORE membrane (22 mm in outer diameter) under the dura, sutured the dura, installed back the removed skull bone with titanium pieces and sutured the scalp. The animal was then sent back to its cage for recovery. Ceftriaxone sodium antibiotic (Youcare Pharmaceutical Group Co. Ltd., China) was administered for one week. We performed a second surgery 45 days later to implant the head posts and imaging window. Three head posts in total were implanted on the animal's skull, with two of them on the forehead and one on the back of the skull. A T-shaped steel frame was fixed to these head posts for head stabilization during training and experiments. Also during this surgery, the skull was re-opened and the subjacent dura was cut off (diameter 16 mm) to exposure the brain cortex. A glass cover-slip (19 mm in diameter and 0.17 mm in thickness) with a titanium ring (12 mm in outer diameter and 10 mm in inner diameter) glued on it was inserted and gently pushed down onto the cortical surface. The titanium ring was glued to the dura and skull using dental acrylic, to establish an imaging chamber. The whole chamber, reinforced by additional thick dental acrylic, was then covered by a steel shell to protect the cover-slip when the animal was returned to the home cage. See Li et al.[30] for further details.

**Behavioral task.** After 10 days of recovery from the second surgery, we trained each monkey to sit in a primate chair with its head restrained, and to perform a fixation task, where keeping fixation on a small white spot (0.1°) within a 2° window for over 2.5 s resulted in a juice reward. Eye position was monitored with an infrared eye-tracking system (ISCAN ETL-200, ISCAN Inc. USA) at 120 Hz. Whenever the eye fixated within a scope of 2° of the center white point, a trial started.

**Visual stimuli.** Visual stimuli were generated using a ViSaGe system (Cambridge Research Systems) and presented on a 17-inch LCD monitor (Acer V173, 80 Hz refresh rate and 1280 × 960-pixel resolution), positioned 45 cm away from the animal's eyes. A small drifting grating (full contrast square waves, four cycle per degree spatial frequency, three cycle per second temporal frequency, 0.2° in diameter) was first used to estimate the RF position of each recorded neuron. For synaptic input tuning measurements (Fig. 1), a set of 81 stimuli were used, which included 72 drifting square-wave gratings (12 orientations, 2 moving directions at 3 spatial frequencies of 1, 2, and 4 cycle per degree) and 9 color patches (red, orange, yellow, green, cyan, blue, purple, white, and black), 1° in size. Direction of motion was not considered in this study, and responses across opposing drift directions were averaged. Visual stimuli were displayed in random order.

We tested both equiluminant colors (8.2 candela per m²) and full saturated colors. The background luminance was set to 25 candela per m² for all our experiments. Full saturated colors were generated by setting the values of the RGB channel to [255, 0, 0]—red, [255, 127, 0]—orange, [255, 255, 0]—yellow, [0, 255, 0] —green, [0, 255, 255]—cyan, [0, 0, 255]—blue, [255, 0, 255]—purple, [255, 255, 255]—white, and [0, 0, 0]—black. We then measured the chromaticity and

luminance of each saturated color using SpectroCAL MKII Spectroradiometer (Cambridge Research System). Accordingly, we set all equiluminant colors to match the blue channel, which had the weakest luminance value at 8.2 candela per m². To calibrate the other equiluminant colors, we adjusted the RGB channel values for each color to a luminance intensity as 8.2 candela per m², while maintaining chromatic saturation of each color channel.

Each stimulus was presented for 0.5 s, with an inter-stimulus interval of 0.5 s and repeated 20 times. For RF mapping (Figs. 4 and 5), we used a set of white/black dot stimuli (0.2–0.3° squares in 5 by 5 position array for basal dendritic inputs, and 0.3–0.4° in 7 by 7 position array for apical dendritic inputs). Each condition was repeated 40 times during RF mapping. To precisely measure the time information of synaptic inputs in experiments with high-imaging speed, 4–6 preferred stimuli were first chosen for each dendritic branch according to previous functional characterizing experiments. Each stimulus was presented for 100 ms with an inter-trial interval of 100 ms and repeated for ~400 times.

**Two-photon imaging.** We performed in vivo two-photon imaging using a Prairie Ultima IV two-photon microscope (Bruker Nano, Inc., formerly Prairie Technologies) and a Ti: Sapphire laser (Mai Tai eHP, Spectra Physics). A 25× objective lens (1.1 N.A., Olympus) with zoom 4× scanning model was used for high-resolution dendritic imaging, which covered a field of view of 136 × 136 μm. Fast resonant scanning mode (512 by 512-pixel resolution, up to 30 frames per second and averaging every four frames) was used in all functional characterization experiments. An even faster scanning mode (64 by 64 pixel at 226 frames per second) while zoomed 8× achieved high temporal resolution imaging (without frame averaging, Fig. 5e). To obtain the orientation map, we imaged with a 16× objective lens (0.8-N.A., Nikon) under zoom 1×, covering a field of view of 850 × 850 μm. We collected 23 neurons in total from three monkeys, sampling neurons with both spiny and smooth dendrites. As we used the hSyn promoter, sampled neurons were either excitatory or inhibitory neurons. The only criteria we used in picking neurons was that the dendritic morphology should be distinct from the background fluorescence.

**Imaging data analysis.** All imaging data were analyzed with custom MATLAB code (Ver. 9.5.0 R2018b, The MathWorks, Natick, MA). To correct the image shifts caused by the relative displacement between the objective and the cortex during the imaging process, we first determined a template image by averaging 1000 frames in the middle of each imaging session and then realigning each frame in a session to the template image, using a normalized cross-correlation-based translation algorithm[30]. After motion correction, we used a newly developed toolbox[38] to extract independent signals on dendrites and determine regions of interest (ROIs). We assumed ROIs as synaptic inputs on neuronal dendrites if they had a 50% intersection with the dendritic domain. To obtain the morphology of each dendritic branch, we first collected an averaged image throughout each experiment, and then filtered this image with a band-pass filter (5 pixels and 50 pixels, 2 orders, respectively). With this processed image, we identified the dendritic domain with PS (Adobe Photoshop CC Ver. 19.1.6) and determined the contour of the dendrites using the software's magic stick tool (tolerance ranging from 10 to 50 in different conditions to optimize fit). Responses of synaptic inputs were computed as the ratio of fluorescence change ($\Delta F/F_0$), where $\Delta F = F - F_0$ and $F_0$ is the baseline fluorescence intensity during the blank screen before stimulus onset in each trial, and $F$ is the fluorescence during stimulus presentation.

To minimize contamination from back-propagation action potentials (BAPs) while Ca-imaging in dendrites from GCaMP6s, we implemented a subtraction procedure (Supplementary Fig. 7). First, a region covering the parent dendritic shaft was circled to measure global BAP induced fluorescence. Second, a mean differential image "diffIm" across all stimulus conditions was calculated and set as a reference for the overall response pattern, obtaining the mean fluorescence intensity of the parent dendritic shaft, "mfi". Third, we computed a differential image "diffIm_1" for each stimulus condition and determined the mean fluorescence intensity for each dendritic region, "mfi_1". Finally, we subtracted the global BAP fluorescence from target dendrites: diffIm_1 – diffIm*mfi_1/mfi.

**SNR maps of iGluSnFR fluorescence signals.** For each pixel on SNR map, an orientation SNR index was defined as the ratio of fluorescence variance between averaged ON and OFF frames (four frames before and during stimulus onsets separately) of 12 orientations under the preferred spatial frequency, and a color SNR index was determined as the ratio of fluorescence variance between averaged ON and OFF frames of nine colors. The SNR value of each pixel was then set as the maximum between the corresponding orientation SNR index and color SNR index.

**Orientation-selective and color-selective inputs.** To collect orientation-selective inputs, the signal-to-noise ratio (SNR) of single synaptic inputs was defined as the ratio of fluorescence variance between the averaged ON and OFF frames of 12 orientations under the preferred spatial frequency:

$$\text{SNR} = \frac{\sigma^2\left(\text{fluo\_ON}_{\text{pref\_SF}}\right)}{\sigma^2\left(\text{fluo\_OFF}_{\text{pref\_SF}}\right)}$$

in which fluo_ON and fluo_OFF are 12 mean fluorescence intensities of target ROI across trials for each orientation condition. Synaptic inputs whose SNR value did not exceed 1 were not included in the analysis. Then, we conducted a balanced one-way analysis of variance (ANOVA) on single-trial response values of 12 orientations under the preferred spatial frequency and set a criteria of $p < 0.05$ to obtain inputs with significant difference in response strength across the 12 orientations. Finally, to measure orientation tuning, tuning curves with step sizes of 15° were fit with the circular Gaussian function:[39]

$$F_G = a \cdot e^{-k \cdot (\cos(\theta - \theta_0) - 1)}$$

where $a$ is the amplitude, $\theta_0$ is the maximum angles and $k$ is the width parameter. Based on the fitted curves, the orientation selectivity index (OSI)[40,41] of individual synaptic input was calculated as:

$$OSI = \frac{Rsp_{max} - Rsp_{ortho}}{Rsp_{max} + Rsp_{ortho}}$$

in which $Rsp_{max}$ is the maximum response strength in the optimal orientation and $Rsp_{ortho}$ represents the response value in the orthogonal orientation (optimal orientation ±90°). Synaptic inputs with OSI exceeding 0.3 were then assigned as orientation-selective inputs.

To determine the color selectivity of each synaptic input, a color selectivity index (CSI)[42] was calculated as:

$$CSI = 1 - \frac{Rsp_{min}}{Rsp_{max}}$$

in which $Rsp_{max}$ and $Rsp_{min}$ are the maximum and minimum response strength to color patch stimuli. Also, a signal-to-noise ratio (SNR) index of single synaptic inputs was defined as the ratio of fluorescence variance between averaged ON and OFF frames of all colors. Moreover, we conducted a balanced one-way analysis of variance (ANOVA) on single-trial response values to obtain inputs with significant differences in response strength across various colors. When the CSI was larger than 0.8, SNR was higher than 1 and response variation across colors was significant (ANOVA, $p < 0.05$), we assigned these synaptic inputs as color-selective.

For each single ROIs, the orientation vs. color index was defined as:

$$\frac{max\_Rsp_{Orien} - max\_Rsp_{Color}}{max\_Rsp_{Orien} + max\_Rsp_{Color}}$$

in which $max\_Rsp_{Orien}$ and $max\_Rsp_{Color}$ are the maximum response intensity to orientation and color respectively. The color vs. orientation index was defined as the negative value of the orientation vs. color index.

**Spatial frequency tuning**. For each orientation-selective input, responses to three different spatial frequencies were fit using the Gaussian function. The preferred spatial frequency for each input was the one corresponding to the maximum of the fitted tuning curve.

**Orientation map**. To demonstrate the spatial organization of orientation preference for the cortical areas we imaged, we first generated raw pseudo-color-orientation maps to check the data quality manually. Differential images from the 12 orientations were averaged across three spatial frequencies and combined to create a raw orientation map in which color denoted orientation preference and brightness represented relative response intensity. To acquire pinwheel-like orientation maps, the differential images for each orientation were first spatially filtered with a 166-µm (corresponding to 100 pixels) low-pass Gaussian filter. The orientation preference of each pixel on the orientation map was then deduced from the vector sum of its response intensities under 12 orientations:[43]

$$Rsp = \sum_{k=1}^{12} rsp_k \exp(2i\theta_k)$$

where $\theta_k$ are the orientations in radians and $rsp_k$ are the response intensities. Each single pixel in the orientation map was then assigned to one of 360 bins according to its orientation preference, marked by different colors (Supplementary Fig. 4). In addition, we drew pixel-by-pixel orientation maps to give a first glance of the raw orientation preference without smoothing, which also demonstrate a structured spatial clustering in consistence with the smoothed patterns. The recording depths for each imaging site were all set to around 200 µm. Then, each neuronal position was aligned to a corresponding imaging plane along the z-axis, and local maps for each target neuron were created.

We then determined the local orientation gradient with the following procedure. We binned pixels into groups of 8 and averaged the orientation preference (vector average) for the binned pixel. The gradient of each pixel was calculated along the x- and y-axis, and set as the square root of the sum of squares along these two axes. This matrix of gradient was then resized (scaled up eight times) and filtered by a Gaussian filter (100 pixels) to generate the final gradient map.

**Blobs and interblob maps**. To estimate the positions of blob and interblob regions[44], we first averaged differential images across high-SF drifting grating

stimulus conditions, and obtained orientation selectivity maps "Orien_diffIm". By averaging differential images across seven color stimulus conditions, we obtained color selectivity patterns "Color_diffIm". Then, we subtracted Orien_diffIm from Color_diffIm and filtered the resultant differential image with a Gaussian filter (25 pixels) to generate color-dominant regions. Those regions having a value larger than 0 were identified as blobs, whereas the remaining regions were identified as interblobs.

**RF maps**. Individual white (125 candela per m²) and black (<0.1 candela per m²) squares were presented randomly at one of 25 (5 by 5 matrix) or 49 (7 by 7 matrix) positions, covering a total area ranging from $0.8 \times 0.8°$ to $2.6 \times 2.6°$ depending on retinotopic eccentricity of target imaging area. Responses to each single square stimulus were recorded and used to define the ON and OFF subfields of spatial RFs. The $5 \times 5$ or $7 \times 7$ response matrix was zoomed in 60 or 40 times, respectively, and spatially filtered with a Gaussian filter (40 or 26 pixels). Next, a colormap changing from blue to green to red was used to denote ON/OFF signals (Figs. 4, 5). ON RF size was defined as the size of areas in an ON RF map with response values higher than a half of maximum response intensity, similarly for OFF RF size. The overall RF size was then defined as the mean value of ON and OFF RF sizes.

**Local clustering of visual features**. Distance-dependent correlations of individual response properties were calculated along each single dendritic branch. Synaptic inputs on the same single dendritic branch were first combined into input pairs ($C_n^2$ combinations in total), and then the feature correlation and dendritic distance of each input pair were computed. These input pairs were later allocated into different groups according to their dendritic distance in 2 µm steps, forming distance-dependent feature correlation plots (Fig. 2f). We fit these scatter plots with a negative exponential function:[12]

$$F(d) = a \cdot e^{-d/\lambda} + \beta$$

in which $a$ is the maximum correlation value, $\lambda$ is the spatial length constant and $\beta$ is the correlation baseline. As a control, the synaptic input positions were randomly shuffled along each dendritic branch, and the corresponding distance-dependent correlation was recalculated in 1000 reshuffles. To obtain orientation tuning correlation coefficients, responses to 12 orientations under the preferred spatial frequency among orientation-selective inputs were used. For spatial frequency, the tuning was fit as described above and evenly interpolated with 11 data points, with which spatial frequency tuning correlations were computed. Color tuning correlations were calculated using responses to 9 colors among color-selective inputs. Pixel-to-pixel correlations of RF[11] ($5 \times 5$ or $7 \times 7$ pixel array according to stimuli set) among orientation-selective inputs were used as a measure of RF similarity.

**Mix of orientation-selective and color-selective inputs**. We used a signal-detection approach by creating a receiver operating characteristic (ROC) curve[45] to analyze the spatial intersection of orientation-selective vs. color-selective inputs. Along one specific direction of each dendritic branch, each pixel on a dendrite was located on color-selective ROIs (True positive), on orientation-selective ROIs (False positive), or on neither (True negative and False negative). In ROC curves for each single neuron (Supplementary Fig. 6), the horizontal axis represented the ratio of cumulative orientation input area on a current point (each point in the red line) to the overall orientation input area, and the vertical axis marked the corresponding ratio of color inputs. The closer the ROC curve was to the unity line, the more interlaced the orientation-selective inputs were with respect to the color-selective inputs. Any obvious spatial clustering of orientation-selective or color-selective inputs on dendrites would lead to a deviation of ROC curve away from the unity line, causing the AUC to deviate from the value of 0.5.

**High-resolution temporal tuning**. Several preferred stimuli for each dendritic branch were used in experiments with high-imaging speed, up to 226 Hz. For each synaptic input, the mean fluorescence intensity through ten frames prior to the stimulus onset across trials was assigned as the fluorescence baseline. To get an averaged time curve of the overall inputs on one dendritic branch, temporal tunings of each synaptic input under its optimal stimulus with maximum response intensity were weighted summed by each ROI's area size. Time curves of apical and basal dendrites were then normalized according to the peak response value and fit with a fourth-order Fourier function (Fig. 5e). The time point corresponding to the half-height of each fitted time curve was collected for comparison of temporal information.

**Statistics**. Sampling neuron sizes and synaptic input numbers are like those in other studies in this field, and the precise details are described in the main text. The degree of correlation was calculated as Pearson's linear correlation coefficient, unless otherwise specified. A ROC analysis was conducted to illustrate the spatial interlacing relationship of orientation-selective and color-selective inputs. The linear dependence between two distinct variables was characterized by the $p$-value or $R^2$ from linear regression analysis (Figs. 2–4 and Supplementary Figs. 2, 5, 8).

The experimenter was blind to each neuron's location when measuring the orientation column structure. No estimates of statistical power were executed before the experiments.

**Reporting summary**. Further information on research design is available in the Nature Research Reporting Summary linked to this article.

## Data availability
The data that support the findings of this study are available from the corresponding author upon reasonable request.

## Code availability
Custom code that support the findings of this study are available from the corresponding author upon reasonable request.

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

## Acknowledgements
We would like to thank Loren L. Looger for the early provision of AAV-iGluSnFR and the Peking University Laboratory Animal Center for excellent animal care. This work was supported by National Natural Science Foundation of China (grant no. 31730109), National Basic Research Program of China (grant no. 2017YFA0105201), National Natural Science Foundation of China Outstanding Young Researcher Award (grant no. 30525016) and a Project 985 grant of Peking University, Beijing Municipal Commission of Science and Technology (grant no. Z151100000915070), and a U.S. National Science Foundation grant to S.L.M. and S.M.C. (1734887).

## Author contributions
N.J. and S.T. designed the study, performed experiments and analyzed data. Y.L., F.L., and H.J. performed surgical procedures. N.J., S.T., S.L.M. and S.M.C contributed to the interpreted results and wrote the paper. S.T. supervised the project.

## Competing interests
The authors declare no competing interests.
