## [Peer Review File · Nature Communications]

Reviewers' Comments:

Reviewer #1:

Remarks to the Author:

This paper measured the spatial and functional distribution of excitatory inputs to neurons in superficial V1. Applying this technique to the primate is extremely impressive and allows for one to address how tuning arises in the most relevant animal model to human visual perception. There is sufficient data from this challenging technique. My first concern is that they did not give enough attention to signal extraction from their images. The off line processing is not a trivial endeavor, but I feel is necessary to make sense of this impressive data set. Additionally, I have major concerns about their color stimulus paradigm and the conclusions that can be drawn with respect to the segregation of form and color selectivity. I think this paper would be of better service if more attention was instead paid to a rigorous signal extraction from the movies.

Major concerns:

1) Signal extraction

My main question is whether the ROI signals are independent from each other and the background. It seems challenging to tease these apart since the ROIs are very small - even the smallest of smearing artifacts (such as brain motion in the Z-axis) is going to mix the signals. Furthermore, the continuity of the dendritic tree in each (averaged) image implies either a wide depth-of-field and/or Z motion. Although perhaps challenging, a more rigorous method of signal processing is warranted for this valuable data set. I have attempted to provide examples of how I think this could be improved, but the authors could certainly have better ideas.

A) Can you apply a more rigorous method to extract independent signals on the dendrite, such as ICA? Perhaps they could apply the pipeline in this paper:
Pnevmatikakis et al. Simultaneous Denoising, Deconvolution, and Demixing of Calcium imaging data.

B) As it stands, the background signal ("local map") is very mysterious. I would like, at the very least, to have a better understanding of it. If it is very weak, it seems that the local maps could mostly just be a heavily smeared version of the "output neuron". On the other hand, if it is very strong, that creates a greater concern that it is bleeding into their ROIs.

How weak or strong is this background signal? Perhaps they could start by just showing an image of SNR... do the dendrites "pop out". Also, what does the orientation map look like without the smoothing?... is it mostly just noise outside of the ROIs? Showing this for every neuron would be most useful.

Also, perhaps they could, in words, speculate on the physiological origin of this background signal.

2) Color vs. orientation dichotomy

I have issues with the interpretation of one of their main results. The conclusion from Fig. 2 is a dichotomy between the different inputs: inputs are either selective for orientation or color. I don't think its possible to make this claim without colored gratings. To summarize, I think its quite possible that they are instead (or in addition) looking at a difference in low-pass vs. band pass spatial frequency tuning - some are tuned for low pass (full-field), whereas others are tuned to bandpass (gratings). If a cell is low-pass for spatial frequency, it will have weak orientation selectivity, but still respond to one of the colors. For example, the orientation selective inputs may be selective to one of their colors, but the full-field stimuli elicit almost no response. In addition, some of the color selective inputs could be orientation selective, but not to monochromatic gratings.

3) Other questions about the color paradigm

-How was color selectivity computed?

-More detail in the methods is warranted regarding how each color was generated on the monitor. e.g. what was the background luminance? Presumably it wasn't 8cd/m².

-Its hard to believe that they got the same results for the saturated and unsaturated case since the contrasts will be dramatically different. They should elaborate on what they mean by "did not vary significantly". Does the presented data come from one or the other, or some combination?

4) Interpretation of a hierarchy

The authors interpret the broad, yet independent functionality of inputs in terms of a hierarchy. They are recording from the upper margin of layer 2/3, which receives input from the bottom of 2/3, not 4C. It seems odd that functional segregation would remain so rigid until this stage. Instead, as described in '2' above, it may not be a segregation of color vs orientation selective inputs, but instead low-pass vs. bandpass inputs which are two sides of a continuum.

Reviewer #2:

Remarks to the Author:

Manuscript by Ju et al., titled "Spatiotemporal functional organization of excitatory synaptic inputs onto macaque V1 neurons" describes spatial distribution patterns of functionally characterized synaptic inputs over the dendritic arbors of V1 superficial layer neurons in Macaques. The authors performed in vivo dendritic input mapping, using two-photon imaging of recently developed genetically-encoded glutamate sensor, iGluSnFR, while the awake monkeys viewed visual stimuli with various features. They report that the majority of the visually-responsive hot spots on the dendrites was either orientation-selective or color-selective and rarely both, and that there was local clustering of hot spots responding to similar stimulus feature. Moreover, differential response properties such as receptive field size and response latencies were observed between apical vs basal dendrites, indicative of differential roles for these distinct groups of dendrites in sensory integration. As far as I am aware, this is the first functional input mapping study performed on a primate with neural activity indicators of any kind, and would be of interest to a broad audience in neuroscience community. However, several key points remain questionable that need to be addressed, including the type of neurons that were being imaged. The nature of glutamate signals that were measured also needs to be clarified and addressed as some of the known kinetics of the indicator may render it unsuitable for precise input mapping in vivo let alone at the resolution scale used by the authors. Specific comments both major and minor are as listed below:

Major comments:

1) The identities of the neurons imaged in this study:

The type of cells imaged in the study is only vaguely described as "superficial layer neurons" and, as the authors state in the text, includes both neural types with spiny and smooth dendrites. This is problematic as the former is likely excitatory neurons while the other could be inhibitory neurons. These two opposing neuronal types should not be pooled together when investigating the functional organization of synaptic inputs over a dendritic arbor as they are known to have very different response properties to begin with. The authors report analyzing a total of 23 neurons from 3 different monkeys? Of these, how many neurons were putative excitatory (spiny) vs inhibitory (smooth) neurons? It would've been insightful if post hoc immunohistochemistry was performed to determine the identities of these neurons, but at the very least, the data should be separated into these two groups before analyzing features such as local clustering of preferred stimulus features. Interneurons in other species have been shown to have longer stretches of the dendritic shafts responding to similar stimulus features (10-20um) while the overall response properties may be heterogenous (Chen et al., Nature 2013) which could no doubt influence the synaptic clustering analyses as shown in Figure 2f.

2) The use of iGluSnFR and dendritic shaft imaging for input mapping:

The authors used a glutamate sensor, iGluSnFR, and all the measurements were made from the “hot spots” on the dendritic shafts rather than the dendritic spines themselves. While this may be less than ideal but acceptable approach for looking at more general organizational patterns of synaptic inputs, this particular reporter is not suitable for investigating local clustering of inputs due to its high affinity to glutamate which makes it even sensitive to the small changes in the glutamate concentration outside of the synaptic cleft like glutamate spillover. The reported $1/e^2$ decay distance for GluSnFR is $\sim 3.6\mu\text{m}$ (Kazemipour et al., Nature Methods 2019). The clustering of dendritic regions responding to similar stimulus features reported in this study is well within this distance, meaning that the reporter is unlikely to have the resolving power to support their findings. This, combined with the point made under Comment 1, leaves the authors conclusions regarding the synaptic clustering weak and unsupported.

3) Regarding functional trade-off between orientation/color inputs and the specificity of the glutamate signals:

The reported trade-off between the two stimulus features is intriguing, but I am having difficulty imagining how this could be especially on apical dendrites which supposedly receive lateral inputs from other cortical neurons such as another superficial layer neurons that are tuned to both color and orientation. Can the authors provide plausible explanations for how this could be?

Rather, given the observation reported in this study that “neurons in iso-domain of orientation column....tended to receive more oriented inputs” rather than color inputs, is it possible that the iGluSnFR is sensing not only glutamate release from those inputs synapsing onto the target neuron but also the spillovers from other axons in the vicinity synapsing onto non-target (unlabeled) nearby neurons (meaning not all the signals detected on the imaged neuron are bona fide synaptic inputs)? Glutamate sensing is tricky because the sensor is membrane bound and reports the level of glutamate OUTSIDE the cell and the more sensitive the reporter is, the more likely it is to pick up non-synaptic glutamate which would be abundant in vivo, especially in iso-domain part of the orientation column stimulated by the preferred orientation stimulus.

Minor comments:

Line 48: “The temporal sequencing of dendritic inputs will lead to new and critical insights” – This statement needs to be clarified as I am not sure what is meant by it.

Line 78-81: “...our data showed that summed input orientation preferences matched the orientation preference of the integrate somal Ca response...was moderately broader than somatic orientation selectivity.” – Was is the case for both neurons residing within the iso-domain and for those in the pinwheel centers?

Line 191: “...one local pair of synaptic inputs might share similar orientation preferences while possessing quite dissimilar RFs” – Previous studies have discovered OFF receptive field-anchoring in which neurons in an orientation column tend to have OFF subregions that are aligned in visual space and ON subregions that are displaced. If you analyzed ON and OFF subregions separately rather than simply using averaged RF centers (Fig 3d, e), do any additional patterns emerge?

Figure4: If you analyzed ON and OFF subregions separately, do you see any differential spatial distribution patterns of each subregions between apical vs basal dendrites?

Figure4d left: As I commented above, I don't think the iGluSnFR signal has the spatial resolution of $<4\mu\text{m}$ (the first two points).

Some of the figure panels may be cited out of order.

Reviewer #3:

Remarks to the Author:

Reviewer comments on manuscript NCOMMS-19-24867

AAV mediated transgene expression and two-photon imaging of living brain in non-human primates proved to be a huge challenge in modern neuroscience in the past decade. However, by stable GCaMP6f expression and two-photon calcium imaging of V1 in awake macaque, Tang's Lab become the first to report functional microarchitectures of thousands of neurons with single-cell resolution in behaving non-human primates (Li et al., Neuron 2017). Following this preceding seminal work, Ju et al. from the same Lab reported here that they have successfully developed a novel technique combining two-photon imaging with a glutamate-sensing fluorescent reporter, iGluSnFR, which resulting much stronger signals at neuronal dendrites when compared with classical dendritic Ca-imaging. There are three main novelties within the current study in the reviewer's opinion:

1. Ju et al. appears to be the first to apply successfully a newly developed glutamate sensor in awake behaving non-human primates.
2. By applying this newly developed powerful technique, Ju et al. mapped excitatory inputs of different basic visual features (such as orientation, color, spatial frequency) on neuronal dendrites in superficial layers of V1 in awake behaving monkeys, demonstrating that dendritic excitatory inputs of V1 neurons encoding different basic visual features are co-existed and mutually overlapped at the circuits' level.
3. Ju et al., also found that the latency of apical inputs is temporally lagged compared to basal inputs, indicating that apical dendrites receive feedback inputs, a concept in the textbook for a long time. The evidence from Ju et al., could be the first evidence at the dendritic level from awake non-human primates by measuring the latency differences between apical and basal dendritic inputs.

Further significance evaluation:

A very recent publication in Science from Callaway's group with classical GCaMP6f expression and two-photon calcium imaging of V1 neuron soma activities in anesthesia monkeys (Garg AK, Li P, Rashid MS, Callaway EM 2019: Color and orientation are jointly coded and spatially organized in primate primary visual cortex. Science 364:1275-1279), has confirmed at the level of thousands of neurons with single-cell resolution that macaque V1 neurons can be both color and orientation tuned (e.g. Livingstone & Hubel, 1988; Leventhal et al. 1995; Johnson et al, 2001; Friedman et al., 2003; Economides et al., 2011). The nice results from Callaway's Lab suggest that orientation and color in V1 are mutually processed by overlapping circuits. The results from Ju et al. reported here in awake monkeys (the point 2 highlighted above) provide direct evidence by visualizing detailed neuronal dendritic excitatory inputs to support the long-existed concept as orientation and color in V1 are mutually processed by overlapping circuits. Therefore Ju et al. work represents a much-advanced novel approach as the first successful application of a newly developed glutamate sensor to record multiple excitatory dendritic inputs of V1 neurons in behaving non-human primates. This newly developed powerful technique for mapping excitatory dendritic inputs encoding different functional features will inevitably open a window for visualizing and probing neuronal integration at a large scale of cortical dendritic inputs' levels underlying many primate sensory and cognitive functions.

In short, I highly recommend this technically much advanced work to be published in Nature Communication for its novelties and significances. However, the authors need to do significant revision in its presentation of their work. Here are some of my major and minor comments and suggestions:

Major comments and suggestions:

Ju et al. in awake monkeys seems to be a parallel work to Garg et al., Science 2019, both using two-photo imaging to probe V1 orientation and color processing mechanisms, one at dendritic inputs level in awake monkey, the other at soma activity level in anesthesia monkeys. I derive this assumption because my impression is that the current format of Ju et al. appears to be more or less in the format of Science but wrongly submitted to Nature Communication. It must be a rejection from the previous submission to Science and resubmitted to Nature Communication with least change or revision. Also Ju et al., did not cite or even mentioned Garg et al., Science 2019 in their manuscript, suggesting that it previously submitted to Science most likely at a similar time period. My biggest concern thus is that Nothing to be shamed if being rejected by Science, instead, Ju et al. should cite and compare their results with the latest published results from Garg et al., Science 2019 as these two elegant works broadly focusing on the same topic but in complementary approaches. In fact, Ju et al. directly confirmed the assumption suggested from Garg et al., Science 2019 with a novel powerful approach in awake monkeys, suggesting a possible global computational integration mechanism that feature selectivity of neurons may result from the sum of all dendritic inputs from local and long-range neuronal circuits.

I would like to suggest that:

1. Combine the result section two and three into a unified section, focusing on the demonstration of the co-organized sparse and clustered dendritic inputs encoding different visual features. These results suggest that orientation, color and spatial frequency could be mutually processed at the same time when presenting combined visual stimuli.
2. The sparse and clustered dendritic excitatory inputs encoding different visual cues are not necessarily competing each other, as the author claimed a trade-off mechanism. Instead, these dendritic excitatory inputs most like facilitates with each other (such as for orientation and color) when using combined visual stimuli, which exclusively happen in the natural environment (for example using colored oriented grating stimuli in a Lab setup instead, as employed by Garg et al., Science 2019). This is the actual facilitative computational mechanism at the dendritic level directly visualized by the authors in the current study.
3. They mapped the receptive field (RF) of dendritic inputs and found that the RF position varied across different dendritic input of same neuron. A previous cat area 17 study has also measured subthreshold RF inputs by in vivo intracellular recording, and they reported that size of subthreshold synaptic inputs (integration visual field) was about four times as large as the minimal discharge field (Bringuier et al., Science 1999). Due to technical limitation, it would be impossible to map the RFs of all the input to individual, but this work gives an idea of subthreshold RF mapping on dendrites. Any possibility to get a large integration visual field based on the integration of local dendritic RFs over large cortical distance?
4. Furthermore, this study compared the excitatory dendritic inputs on apical dendrites in upper layer with those on basal dendrites in the deeper layer. They found the apical dendritic inputs have similar but broader orientation preference with the basal dendritic inputs, and the apical dendritic inputs have larger RF size and lagged latency compared with basal dendritic inputs in the same neuron. It provides the first visualized evidence of temporal difference between dendrites in different depth. However, whether the recorded inputs on apical dendrites representing feedback signal and that on basal dendrite representing feedforward signal was not proved, at least not carefully discussed.
5. This research provides a new, powerful and effective approach which can help understanding the dendritic mechanism of signal processing on the surface of non-human primate cortex. Compared with the research of Garg et al., Science 2019 recording soma activities, this work provides much more details inside the neuronal dendrites and local circuits. However, the presentation of results (including the research background with specific scientific questions, the details of results, the description and discussion) is not clear enough. The manuscript needs major revisions. If it were presented as elegantly as Garg et al., Science 2019, the manuscript might be published already in Science.

Minor moments and suggestions:

1. The English writing is neither smooth nor flowing, because of the short format of the current version (it appears still in its previous Science format). Nature Communication will provide an idea platform for this work to be fully introduced, explained and discussed. The English itself must be clearer and need somebody with native English to improve it carefully and seriously.
2. Which dots representing neuron I, II, III, and IV should be more clearly marked in Figure 2m, maybe also in Figure 2j and 2k.
3. The figure legends should be presented more pithily. Some details of experiments could be moved into Method part, and some description of results could be moved into Result texture.
4. For the histogram plots in Figure 2 h & j, the scale of y axes should be 0~100, rather than 0~1 if it represents percentage (%).
5. In Figure 4f (right), what is the meaning of red circle? It is not referred in the legend nor in the figure, does it represent the sampled neuron?
6. The manuscript title could be changed to something like "From local to global: The integrated local dendritic inputs determine neural functional selectivity in primate V1", which depicts what the authors have observed by applying their newly develop powerful technique to awake monkey V1.

Responses to Reviewers' comments:

We thank the reviewers for their kind and thoughtful critiques. All three reviewers recognized the potential significance of our contributions while bringing up several specific technical concerns. We addressed all of the reviewers' advice and revised the manuscript extensively to elucidate our findings more fully. Below we provide point-to-point responses to the reviewers' comments.

Reviewer #1 (Remarks to the Author): This paper measured the spatial and functional distribution of excitatory inputs to neurons in superficial V1. Applying this technique to the primate is extremely impressive and allows for one to address how tuning arises in the most relevant animal model to human visual perception. There is sufficient data from this challenging technique. My first concern is that they did not give enough attention to signal extraction from their images. The off line processing is not a trivial endeavor, but I feel is necessary to make sense of this impressive data set. Additionally, I have major concerns about their color stimulus paradigm and the conclusions that can be drawn with respect to the segregation of form and color selectivity. I think this paper would be of better service if more attention was instead paid to a rigorous signal extraction from the movies.

Re.: We thank reviewer #1 for commending the significance and novelty of our work. We have now enhanced the rigor of the signal extraction analyses from the raw data. We have now also provided further description of how we segregated form vs color selectivity tuning from the data. Details below.

Major concerns:

1) Signal extraction

My main question is whether the ROI signals are independent from each other and the background. It seems challenging to tease these apart since the ROIs are very small - even the smallest of smearing artifacts (such as brain motion in the Z-axis) is going to mix the signals. Furthermore, the continuity of the dendritic tree in each (averaged) image implies either a wide depth-of-field and/or Z motion. Although perhaps challenging, a more rigorous method of signal processing is warranted for this valuable data set. I have attempted to provide examples of how I think this could be improved, but

the authors could certainly have better ideas.

Re.: In our experiments, individual inputs showed significant tuning to orientation, spatial frequency or color, as shown in Figure 1C-D. Given that the stimuli were presented randomly, and taking into account the high repeatability of the single-trial fluorescence traces (Supplementary Fig. 2), our data indicate that the signals we recorded did not arise from brain motion related changes, which we rule out given that such artifacts cannot explain the selectivity to visual stimulus features found in our data.

We do agree that spatial resolution is important for high quality sampling of spine data, and we here show that our imaging resolution is sufficiently high to tease apart different ROIs. Our two-photon microscope (Bruker Nano, Inc., Prairie Ultima IV two-photon microscope) has been calibrated to capture high resolution two-photon images: The point-spread function (PSF) has been measured at: $0.4\mu\text{m}$ X-Y plane resolution, and $2.0\mu\text{m}$ Z axis, when using the 25x Olympus objective used for these experiments. This is a typical range of measurements (compared to other studies in the field), and we have achieved the same diffraction-limited microscopic resolution—near the physical limit also found in other similar two-photon systems—reported in prior work employing the imaging of dendrites, spines, and other filaments and small cells in the neuropil. In our dendritic imaging experiments, the sampling resolution was set to about 4 pixels/ μm . This setting oversampled to magnify beyond the physical limitation of the two-photon imaging system, which is a standard approach. With this approach, single-ROI signals were robust and repeatable (Figure 1C, 1E and S2). We also replicated these results using a higher power objective combined with higher magnification to achieve higher spatial resolution (18 pixels/ μm), using a $60\times$ objective lens under zoom $8\times$ and 512×512 pixels (Figure R1). Because those recordings produced similar results, we conclude that the results with the 25X objective have the necessary precision for spine analysis, and thus represent accurate and valid results.

Figure R1. Left, an averaged image obtained with a 60 \times objective lens under zoom 8 \times , covering a field of view of 28 μm , averaged across 1000 frames, recorded at a depth of 192 μm . Right, the differential image (ON-OFF) under one visual stimulus exhibited hotspots on target dendrites.

As for the concern about potential z-axis fluorescence leaks across different depths, we can rule out the possibility that our data is systematically contaminated by out-of-plane recording artifacts because of the fine optical sectioning obtained in two-photon imaging (a function of the PSF along Z axis $\sim 2.0\mu\text{m}$ for our system). Figure R2 shows a series of continuous Z-stack images, with each pair of neighboring images having a depth difference of $3\mu\text{m}$: these images vary greatly and thus are not mixing beyond the z-axis diffraction-limited resolution of our 25X objective. This control experiment supplies a direct impression of fine imaging resolution along the Z axis. To obtain additional dendritic inputs in one experimental session, we always chose a target focal plane where longer continuous dendritic trees existed (as an example, we set the imaging depth at $127\mu\text{m}$ for the neuron in Figure R2). Hence, the continuity of the dendritic tree arises from the intrinsic neuronal morphology and from the imaging plane we chose, rather than from a wide depth-of-field or from Z motion. As with all two-photon recordings, fluorescence and signal crosstalk may have occasionally occurred across different depths, but the likelihood that this caused a regular or systematic contamination of our results is very low, and it would mean that there were hidden invisible dendrites aligned with our dendrites, that we could not see in our Z-stacks. We do not believe that this could have occurred in any systematic fashion that we could miss, especially as it would mean that these hidden dendrites nevertheless are somehow contributing a fluorescence artifact significant enough to influence our results. Therefore, such a possibility is vanishingly low given the outcome of our controls. As also shown in Supplementary Fig. 2 c-g, the background fluorescence change was minute compared to the robust signals we found on our dendritic recordings. Additionally, regions

that exhibited robust fluorescence in our analyses arose in positions matching our dendritic branches (Figure R1, Supplementary Fig. 2 b-c, Video 1 and 2), arguing against the possibility that our signals originated in planes that were out of focus.

Figure R2: Averaged images obtained with a 25x objective lens under zoom 1x, covering a field of view of 128 μ m, averaged across 100 frames each, recorded at a 3 μ m depth interval from 120 μ m to 132 μ m.

We nevertheless agree that our study would greatly benefit from more rigorous signal processing, and we have incorporated such methods and analyses in our revised MS (please see the corresponding subheading below).

A) Can you apply a more rigorous method to extract independent signals on the dendrite, such as ICA? Perhaps they could apply the pipeline in this paper:

Pnevmatikakis et al. Simultaneous Denoising, Deconvolution, and Demixing of Calcium imaging data.

Re.: Following your kind suggestion, we have now employed Pnevmatikakis et al's extraction procedure for each individual neuron, and re-processed all corresponding statistics as a function of the new ROIs, modifying the relevant figures accordingly in the revised manuscript (Fig. R3). The results continue to support all our previous conclusions, with the exception of the significance level in the previous revision's Fig. 4d, which fell from from $p=0.047$ to $p=0.18$ (Figure R4). We assume this occurred because we changed the method of characterizing ROIs, and thus the ROI population changed; because we continued to exclude those neurons with fewer than 15 ROIs on their Apical Dendrites (ADs), the sampling size of analyzed neurons fell from 9 neurons to 6. Though the tendency of biased distribution still existed, the power value for this t -test equaled 0.21 (with the hypothesis that AD beta value differed significantly from Basal Dendrites (BDs)), indicating a lack of

sampled neurons. We therefore removed this panel (original Fig. 4d) from the manuscript and we will readdress this specific analysis in the future, when there is sufficient power to conduct a proper analysis using Pnevmatikakis et al's method.

Figure R3: ROIs with Pnevmatikakis et al's method to extract independent signals on the dendrites.

Figure R4: The removed panel from original Fig. 4d. Left, relationship between the dendritic distance of ROI pairs and their orientation tuning correlation coefficients on AD (red) and BD (blue) of the example neuron in Fig. 4. β -value represents the correlation baseline (see Methods). Error bar, SEM. Right, AD β -value versus BD's of 6 neurons (neurons with less than 15 ROIs on their AD were discarded). The example neuron is denoted with a red circle.

B) As it stands, the background signal ("local map") is very mysterious. I would like, at the very least, to have a better understanding of it. If it is very weak, it seems that the local maps could mostly just be a heavily smeared version of the "output neuron". On the other hand, if it is very strong, that creates a greater concern that it is bleeding into their ROIs.

How weak or strong is this background signal? Perhaps they could start by just showing an image of

SNR... do the dendrites “pop out”. Also, what does the orientation map look like without the smoothing?... is it mostly just noise outside of the ROIs? Showing this for every neuron would be most useful. Also, perhaps they could, in words, speculate on the physiological origin of this background signal.

Re.: As to the concern about the local orientation maps, the first point we should clarify is that orientation map measurements and dendritic imaging were conducted in separate experiment sessions with distinct apparatus settings, as required by the differing resolutions and imaging conditions for the different kinds of data. Specifically, high resolution dendritic imaging was conducted using a 25× objective lens (1.1 N.A., Olympus) with zoom 4× scanning mode, covering a field of view of 136μm × 136μm. Because a hyper-column in macaque V1 takes about 1mm² of cortical area, we obtained maps of orientation pinwheel structure using a much larger field of view (about 6.25X larger): a 16× objective lens (0.8-N.A., Nikon) with zoom 1×, which covered a field of view of 850μm × 850μm (left panels in Figure R5). Smoothed orientation maps of these recording sites show clear orientation pinwheel structure (right panels in Figure R5). In addition, as the reviewer requests, here we draw pixel-by-pixel orientation maps to give a first glance of the raw orientation preference without smoothing (middle panels in Figure R5), which also demonstrate a structured spatial clustering in consistence with the smoothed patterns. The recording depths for each imaging site were all set to around 200μm. Then, each neuronal position was aligned to a corresponding imaging plane along the Z-axis, and local maps for each target neuron were determined. Since individual dendritic branches were relatively tiny at this imaging scale level, local orientation maps were measured from global membrane-located iGluSnFR fluorescence.

Figure R5: Orientation maps for each imaging site from 3 monkeys, recorded with a 16 \times objective lens (0.8-N.A., Nikon) under zoom 1 \times , covering a field of view of 850 $\mu\text{m} \times 850\mu\text{m}$. For each imaging site, left panel: an averaged image from all frames acquired during corresponding recording session; middle: pixel-by-pixel orientation map without smoothing; right: orientation map after Gaussian filtering, black circles represent the cortical position of each target neuron.

As for the concern about relative strength of background signals versus dendritic signals during high resolution dendritic imaging, we found that signals on dendritic branches were much stronger and could be isolated reliably (Supplementary Fig. 2 a-g, see also Video 1 & 2). As the reviewer suggested, we also calculated SNR maps for individual neurons, and signals on dendrites indeed popped out (Figure R6). It is worth noting that a few hot-spots outside target dendritic regions exhibited strong activity during visual stimulation, though such active regions could be reliably excluded from targeted dendritic inputs for each single neuron (Figure R1 A-F), based on their spatial locations.

Figure R6: SNR maps of iGluSnFR fluorescence signals. a Left, two-photon image of an example neuron at a depth of 127 μm , averaged across 1000 frames from one recording session. Right, pixel-by-pixel SNR map of fluorescence signals under the same field of view. For each pixel on this map, an orientation SNR index was defined as the ratio of fluorescence variance between averaged ON and OFF frames of 12 orientations under the preferred spatial frequency, and a color SNR index was determined as the ratio of fluorescence variance between averaged ON and OFF frames of 9 colors. The SNR value of each pixel was then set as the maximum between the corresponding orientation SNR index and color SNR index. SNR value is denoted by the colorbar on the right side. **b** Same as **a**, for another sample neuron at a depth of 105 μm .

2) Color vs. orientation dichotomy

I have issues with the interpretation of one of their main results. The conclusion from Fig. 2 is a dichotomy between the different inputs: inputs are either selective for orientation or color. I don't think its possible to make this claim without colored gratings. To summarize, I think its quite possible that they are instead (or in addition) looking at a difference in low-pass vs. band pass spatial frequency tuning - some are tuned for low pass (full-field), whereas others are tuned to bandpass (gratings). If a cell is low-pass for spatial frequency, it will have weak orientation selectivity, but still respond to one of the colors. For example, the orientation selective inputs may be selective to one of their colors, but the full-field stimuli elicit almost no response. In addition, some of the color selective inputs could be orientation selective, but not to monochromatic gratings.

Re.: Thank you for pointing out that it is less rigorous for us to describe inputs as either orientation- or color-selective without using colored gratings. We have now re-analyzed our data using Pnevmatikakis et al.'s procedure, and we have also introduced an orientation versus color index to better illustrate the functional preference of individual synaptic inputs. We found that the preference between orientation versus color was quantified continuously, and we could avoid creating a false dichotomy unintentionally (Fig. 3b). Furthermore, we should point out that previous studies have found color-orientation feature integration only in Layer 2/3 neurons (Leventhal et al., J

Neurosci, 1995; Friedman et al., J Physiol, 2003; Johnson et al., J Neurosci, 2008), including a recent study that used cellular-resolution two-photon imaging to show that orientation and color are jointly coded by V1 superficial neurons (Garg et al., Science, 2019). Our results are thus consistent with these studies and moreover present direct evidence of dendritic-level computations for orientation versus color processing mechanisms in V1 neurons. Also, following from the updated analysis, a large proportion of inputs tuned to both orientation and color, which makes sense given that the previous research suggests that these inputs most likely come from Layer 2/3.

We agree that we may have potentially underestimated the proportion of inputs selective to both orientation and color in our previous revision. We are also grateful that Reviewer #1 suggested we might instead look at a difference between low-pass versus band-pass spatial frequency tuning. However, without colored grating stimuli, it is also not enough to assign an input as low-pass or band-pass selective, as color and SF are conflated (as the reviewer correctly pointed out). For example, a color-selective input may respond to both full-field color patches and colored gratings but not to monochrome gratings, which would make it both low-pass and band-pass selective, while it would be determined solely as low-pass selective without colored grating stimuli. Considering that orientation-color co-tuned inputs could be regarded as both orientation-selective and color-selective, our main conclusion that nearly all individual neurons received both abundant orientation and color inputs remains valid. Even so, we have now modified our description in the manuscript to avoid misinterpretation.

3) Other questions about the color paradigm

-How was color selectivity computed?

Re.: Thank you for pointing out that we missed a detailed description about color selectivity. To determine the color selectivity of each synaptic input, a selectivity index was calculated as $1 - (\text{minimum response})/(\text{maximum response})$ (Tomoyuki et al., J Neurosci, 2014). Also, a signal-to-noise-ratio (SNR) index of single synaptic inputs was defined as the ratio of fluorescence variance between averaged ON and OFF frames of all colors. Moreover, we conducted a balanced one-way analysis of variance (ANOVA) on single trial response values to obtain inputs with significant differences in response strength across various colors. When the color selectivity index was larger than 0.8, SNR was higher than 1 and response variation across colors was significant

(ANOVA, $p < 0.05$), we assigned these synaptic inputs as color-selective. This is now clarified in the revised manuscript.

-More detail in the methods is warranted regarding how each color was generated on the monitor. e.g. what was the background luminance? Presumably it wasn't 8cd/m².

Re.: Agreed, a more detailed description of color generation is required. The background luminance was set to 25 cd/m² for all our experiments on a 17-inch LCD monitor (Acer V173, 80Hz refresh rate and 1280 × 960-pixel resolution). Full saturated colors were generated by setting the values of the RGB channel to [255, 0, 0] - red, [255, 127, 0] - orange, [255, 255, 0] - yellow, [0, 255, 0] - green, [0, 255, 255] - cyan, [0, 0, 255] - blue, [255, 0, 255] - purple, [255, 255, 255] - white and [0, 0, 0] - black. We then measured the chromaticity and luminance of each saturated color using SpectroCAL MKII Spectroradiometer (Cambridge Research System). Accordingly, we set all equiluminant colors to match the blue channel, which had the weakest luminance value at 8.2 cd/m². To calibrate the other equiluminant colors, we adjusted the RGB channel values for each color to a luminance intensity as 8.2 cd/m², while maintaining chromatic saturation of each color channel. We have now added these methods to the revised manuscript.

-Its hard to believe that they got the same results for the saturated and unsaturated case since the contrasts will be dramatically different. They should elaborate on what they mean by “did not vary significantly”. Does the presented data come from one or the other, or some combination?

Re.: Thank you for pointing out that this conclusion requires further data to support it. We now present our analysis in the revised manuscript (and details below). The data presented in the main figures came from saturated color. At the beginning of our experiment (on Monkey 1 and Monkey 2), we only used saturated colors. However, later we realized that it was necessary to eliminate the effect of luminance in our experiment. We then did control experiments to compare the difference between saturated vs. equiluminant color-evoked dendritic responses (on Monkey 3). Color preference of 79% dendritic inputs stayed stable when comparing saturated vs. equiluminant stimuli (see below in Figure R7b, bottom panel, for error tolerance between neighboring colors, red shadow). Moreover, the averaged response intensity of dendritic inputs did not vary significantly under saturated versus

equiluminant chromatic stimuli ($p > 0.05$ by Wilcoxon rank sum test, Figure R7c, e). This result makes intuitive sense given the fact that, even in our equiluminance condition, which is less bright than the saturated condition, the luminance was nevertheless set at a relatively high luminance because it is the highest luminance setting of the blue channel, which is perceptually quite bright. Thus, we should not expect a significant difference between the saturated and equiluminant conditions.

Figure R7: Comparison between non-equiluminant and equiluminant chromatic stimulus evoked responses. a Peak response intensities of ROIs to equiluminant color stimuli versus non-equiluminant (but colored) stimuli. Red line, linear regression. 7 neurons in total from Monkey 3 were recorded. Bottom, statistics for all ROIs from all 7 neurons. b Color preference of each ROI among equal-luminance colors versus non-equal-luminance colors. Brightness denotes ROI proportion, referring to colorbar above the bottom figure. c Averaged response intensity of all ROIs under non-equiluminant versus equiluminant chromatic stimuli. ROIs in a-c were collected only according to non-equiluminant color stimuli. Asterisk, significant difference, by Wilcoxon rank sum test; $p=0.11$ when combining all ROIs. d Non-equiluminant and equiluminant chromatic stimuli response strengths within ROIs that were tested with both. e, Same as c, statistics of averaged response intensities; $p=0.07$ when combining all ROIs.

4) Interpretation of a hierarchy

The authors interpret the broad, yet independent functionality of inputs in terms of a hierarchy. They are recording from the upper margin of layer 2/3, which receives input from the bottom of 2/3, not 4C. It seems odd that functional segregation would remain so rigid until this stage. Instead, as described in '2' above, it may not be a segregation of color vs orientation selective inputs, but instead low-pass vs. bandpass inputs which are two sides of a continuum.

Re.: We agree with Reviewer #1's comment about a functional hierarchy and that we did not explain it carefully in the original manuscript. Since we have mapped ON/OFF receptive fields for each input, we tried the methods used in 'Kuo-Sheng Lee et al., Nature, 2016' to classify inputs as simple or complex (namely inputs from simple or complex cells), and found that nearly all inputs are complex inputs. Taking into account that neurons in layer 4 are mainly simple cells, and that complex cells appear on layer 2/3, we agree that most of the dendritic inputs we recorded originate from the bottom of layer 2/3 rather than from layer 4C. We also agree that, without using colored grating stimuli, our conclusion of a functional segregation of orientation versus color selective inputs lacked rigor. We have now modified the manuscript to address this limitation. We also want to thank the reviewer for voicing this criticism, as we are now considering a detailed follow-up study about orientation and color segregation using colored grating stimuli.

Reviewer #2 (Remarks to the Author):

Manuscript by Ju et al., titled “Spatiotemporal functional organization of excitatory synaptic inputs onto macaque V1 neurons” describes spatial distribution patterns of functionally characterized synaptic inputs over the dendritic arbors of V1 superficial layer neurons in Macaques. The authors performed in vivo dendritic input mapping, using two-photon imaging of recently developed genetically-encoded glutamate sensor, iGluSnFR, while the awake monkeys viewed visual stimuli with various features. They report that the majority of the visually-responsive hot spots on the dendrites was either orientation-selective or color-selective and rarely both, and that there was local clustering of hot spots responding to similar stimulus feature. Moreover, differential response properties such as receptive field size and response latencies were observed between apical vs basal dendrites, indicative of differential roles for these distinct groups of dendrites in sensory integration. As far as I am aware, this is the first functional input mapping study performed on a primate with neural activity indicators of any kind, and would be of interest to a broad audience in neuroscience community.

However, several key points remain questionable that need to be addressed, including the type of neurons that were being imaged. The nature of glutamate signals that were measured also needs to be clarified and addressed as some of the known kinetics of the indicator may render it unsuitable for precise input mapping in vivo let alone at the resolution scale used by the authors. Specific comments both major and minor are as listed below:

Major comments:

1) The identities of the neurons imaged in this study:

The type of cells imaged in the study is only vaguely described as “superficial layer neurons” and, as the authors state in the text, includes both neural types with spiny and smooth dendrites. This is problematic as the former is likely excitatory neurons while the other could be inhibitory neurons. These two opposing neuronal types should not be pooled together when investigating the functional organization of synaptic inputs over a dendritic arbor as they are known to have very different response properties to begin with. The authors report analyzing a total of 23 neurons from 3 different monkeys? Of these, how many neurons were putative excitatory (spiny) vs inhibitory (smooth) neurons? It would’ve been insightful if post hoc immunohistochemistry was performed to determine the identities of these neurons, but at the very least, the data should be separated into these two groups before analyzing features such as local clustering of preferred stimulus features.

Interneurons in other species have been shown to have longer stretches of the dendritic shafts responding to similar stimulus features (10-20um) while the overall response properties may be heterogenous (Chen et al., Nature 2013) which could no doubt influence the synaptic clustering analyses as shown in Figure 2f.

Re.: Thanks for your suggestion! We agree that it is quite important to identify the neuronal type and conduct data analyses separately for different types of neurons. Reviewer #2's specific concern was the previously reported homogeneous orientation tuning along continuous domains on inhibitory neuron dendrites (Chen et al., Nature, 2013), and how this might influence the synaptic clustering analyses. It is worth noting that, whereas Chen et al. identified signals on spines of excitatory neurons as synaptic inputs, they did not identify equivalent signals from large dendritic domains. Instead, they identified dendritic-spike induced fluorescence changes. This is clarified in the last paragraph of 'Calcium transients in GABAergic dendrites' section in Chen et al., in which they state that the large size of their domains might reflect spatial clustered homogeneous inputs or a few sparse, randomly distributed strong inputs, amplified by local postsynaptic mechanisms. Hence, from Chen et al, the spatial organization of inputs onto inhibitory neurons remains unknown, and the inference that there would be significant difference of spatial clustering properties between inputs onto excitatory versus inhibitory neurons is not clear. We also disagree with this interpretation because Ca-imaging experimenters post-process their data to control for the effects of back-propagating spikes, and this is a large source of noise (see Fig. 5d in Chen's paper for an example, which suggests that even on dendritic spines, the input signals they could obtain were pretty sparse). By contrast, iGluSnFR resulted in strong signals on neuronal dendrites and maintained single-input measuring resolution in the meantime, which tends to be a more reliable sensor for dendritic input imaging. Thus, though we have no quarrel with the Chen et al. data, it is not necessarily true that their results differ from ours, due to the discrepancy in techniques used in each study.

Having said this, we have now done the analysis as Reviewer #2 requested. We classified neurons as either smooth or spiny ones according to their dendritic spine densities (11 neurons were determined as smooth ones and 12 as spiny ones, Figure R8a). Then, by comparing the λ -value (spatial length constant) and β -value (correlation baseline) of spatial tuning correlation between smooth versus spiny neuronal classes (Figure R8 b), no significant difference was observed for

orientation tuning spatial clustering constant between smooth and spiny neurons (λ -value, Figure R8b).

Currently, it remains a major technical challenge to target specific types of neurons in macaque cortex *in vivo* (Dimidschstein *et al.*, *Nat. Neurosci.*, 2016; Nagai *et al.*, *Biochemical and Biophysical Research Communications*, 2019). Due to the technical limitations of genetic selection within macaque cortical neurons, nearly all two-photon imaging studies in this field could only make general conclusions across all types of neurons (Nauhaus *et al.*, *Nat. Neurosci.*, 2012; Sadakane *et al.*, *Cell Report*, 2015; Seidemann *et al.*, *eLife*, 2016; Garg *et al.*, *Science*, 2019). This remains true for our study, and the revised manuscript now provides a more detailed discussion of these factors.

Figure R8: Comparison of functional organization between synaptic inputs onto smooth versus spiny dendrites. a, Spine densities of the recorded neurons. Neurons with dendritic spine density lower than 0.3 units per 10 μm were assigned as smooth ones (putative inhibitory, marked in blue), and those with spine density higher than 0.3 as spiny ones (putative excitatory, marked in red). b, Left, relationship between the dendritic distance of inputs and their orientation tuning correlation coefficients within smooth neurons (top, in blue) and spiny neurons (bottom, in red). Right, statistics (mean value and 95% confidence interval) of λ -value and β -value of dendritic orientation tuning for each kind of neurons.

2) The use of iGluSnFR and dendritic shaft imaging for input mapping:

The authors used a glutamate sensor, iGluSnFR, and all the measurements were made from the “hot spots” on the dendritic shafts rather than the dendritic spines themselves.

While this may be less than ideal but acceptable approach for looking at more general organizational patterns of synaptic inputs, this particular reporter is not suitable for investigating local clustering of inputs due to its high affinity to glutamate which makes it even sensitive to the small changes in the

glutamate concentration outside of the synaptic cleft like glutamate spillover. The reported $1/e^2$ decay distance for GluSnFR is $\sim 3.6\mu\text{m}$ (Kazemipour et al., Nature Methods 2019). The clustering of dendritic regions responding to similar stimulus features reported in this study is well within this distance, meaning that the reporter is unlikely to have the resolving power to support their findings. This, combined with the point made under Comment 1, leaves the authors conclusions regarding the synaptic clustering weak and unsupported.

Re.: Please note that, while the $1/e^2$ decay distance for GluSnFRs was $3.6\mu\text{m}$ as illustrated by Kazemipour et al. (corresponding cross-validation tuning correlation value higher than 0.06), the half-life ($1/2$) decay distance is less than $1\mu\text{m}$ (correlation value higher than 0.22). In other words, whereas iGluSnFR could perceive small amount of glutamate that spread up to $3.6\mu\text{m}$ away, robust detection of glutamate occurred primarily within $1\mu\text{m}$. This is supported by Kazemipour et al.'s work as our results are consistent and we obtained similar sizes for inputs onto neuronal dendrites from our iGluSnFR imaging experiments (the mean value of ROI radius equaled $0.76\mu\text{m}$, Figure R9). Such spatial scale of iGluSnFR signals could then support our clustering analysis. Combined with our reply to Reviewer #2's concern under Comment 1, we conclude that it is indeed reasonable to analyze the spatial clustering in our manuscript given our analyses of Reviewer #2's concerns, which we have now discussed thoroughly in the revised manuscript.

As for the concern about input signal positions on dendrites, we wish to clarify that both our data and Kazemipour et al.'s contain inputs onto both dendritic shafts and spines (see ROI 1 in Fig. 1, see also Fig. 2 b-e and Supplementary Fig. 6a; Fig. 4c in Kazemipour et al.'s work). Compared to traditional Ca-imaging, imaging with glutamate sensor iGluSnFR does have a heightened sensitivity to glutamate inputs as iGluSnFR eliminates the need to post-process the data and control for back-propagating spikes (see Supplementary Fig. 6, also see the visible difference shown in Fig. 4c in Kazemipour et al., Nature Methods, 2019). For dendritic input imaging, it is an advantage rather than a disadvantage to obtain signals on dendritic shafts, since if there are glutamate inputs on shafts, they are excitatory inputs that should be accounted for.

Figure R9: Size statistics for all 1818 ROIs obtained from 3 monkeys. ROI radius was defined as the square root of corresponding ROI area. The mean value of ROI radius was 0.76 μm, the minimum was 0.53 μm and the maximum was 1.17 μm.

3) Regarding functional trade-off between orientation/color inputs and the specificity of the glutamate signals: The reported trade-off between the two stimulus features is intriguing, but I am having difficulty imagining how this could be especially on apical dendrites which supposedly receive lateral inputs from other cortical neurons such as another superficial layer neurons that are tuned to both color and orientation. Can the authors provide plausible explanations for how this could be?

Re.: Thank you for the opportunity to clarify this issue. The conclusion about orientation/color functional trade-off was found only on inputs to basal dendrites, not apical dendrites. It was not until Fig. 4 that inputs onto apical dendrites were analyzed. We have now clarified this point in the revised version.

Rather, given the observation reported in this study that “neurons in iso-domain of orientation column....tended to receive more oriented inputs” rather than color inputs, is it possible that the iGluSnFR is sensing not only glutamate release from those inputs synapsing onto the target neuron but also the spillovers from other axons in the vicinity synapsing onto non-target (unlabeled) nearby neurons (meaning not all the signals detected on the imaged neuron are bona fide synaptic inputs)? Glutamate sensing is tricky because the sensor is membrane bound and reports the level of glutamate OUTSIDE the cell and the more sensitive the reporter is, the more likely it is to pick up non-synaptic glutamate which would be abundant in vivo, especially in iso-domain part of the orientation column stimulated by the preferred orientation stimulus.

Re.: As indicated in our reply to Comment 2, the highly localized iGluSnFR signals (with a mean value of ROI radius equaling 0.76 μm , Figure R9) provide strong evidence for local glutamate release, rather than spillover from nearby synaptic clefts. Representative movies of the iGluSnFR recordings for iso-domain neurons directly show how local iGluSnFR signals integrate in the dendrite (Supplementary Video. 2). Even the strongest signals did not spillover to nearby locations. Because clearly identified inputs do not spillover to other clearly identified nearby locations, even during strong stimulation, it follows that our other recordings are not driven by misattributed spillover.

Minor comments:

Line 48: “The temporal sequencing of dendritic inputs will lead to new and critical insights” – This statement needs to be clarified as I am not sure what is meant by it.

Re.: We agree. Now rectified in the revised manuscript.

Line 78-81: “...our data showed that summed input orientation preferences matched the orientation preference of the integrate somal Ca response...was moderately broader than somatic orientation selectivity.” – Was is the case for both neurons residing within the iso-domain and for those in the pinwheel centers?

Re.: Yes. In Monkey 3, we collected 6 neurons that co-expressed RCaMP and iGluSnFR, and 5 neurons among them were orientation selective (same evaluation criteria as the one we used to classify dendritic inputs). For these neurons, somatic orientation preference matches well with the summed input orientation preference (Figure R10a), and the summed input orientation selectivity was always broader than somatic orientation selectivity regardless of the neuron's relative position in the orientation column structure (Figure R10b).

Figure R10: Functional comparison between RCaMP and iGluSnFR signals. a, Somatic orientation preference versus that of summed dendritic ROI responses for each single neuron. b, Full width at half maximum (FWHM) of somatic orientation tuning (in red) or summed dendritic ROI orientation tuning (in green) versus neural distance to pinwheel center. Black dashed line links data dots from same neuron.

Line 191: “...one local pair of synaptic inputs might share similar orientation preferences while possessing quite dissimilar RFs” – Previous studies have discovered OFF receptive field-anchoring in which neurons in an orientation column tend to have OFF subregions that are aligned in visual space and ON subregions that are displaced. If you analyzed ON and OFF subregions separately rather than simply using averaged RF centers (Fig 3d, e), do any additional patterns emerge?

Re.: The reviewer may refer to the study done by Lee et al., Nature, 2016. We have also tried to analyze ON and OFF subregions separately, however, we found no significant difference between them, possibly due to the fact that most of the synaptic inputs we recorded were classified as complex inputs (using the same classification pipeline as Lee et al. described in their Methods), whereas Lee et al.'s conclusion was based on simple cells. Therefore, our two studies may be examining different levels in the visual processing hierarchy.

Figure4: If you analyzed ON and OFF subregions separately, do you see any differential spatial distribution patterns of each subregions between apical vs basal dendrites?

Re.: We have done such analyses (Figure R11), and found the distribution of ON and OFF RFs more scattered for inputs onto apical dendrites only in 3 neurons out of 9 that we collected in total ($p < 0.05$,

two-sample F-test for equal variances). Our study did not set out to test the Lee et al's hypothesis and it thus does not have the power to make a robust conclusion on the basis of the present data.

Two-sample F-test for equal variances

Figure R11: Spatial distribution of ON and OFF subregions for inputs onto apical (AD) versus basal dendrites (BD). From left to right, these panels represent for: ON/OFF mean RF center distribution (shown as eccentricity in degree), ON RF center distribution, histograms for horizontal (x) and vertical (y) positions of ON RF centers between AD and BD, OFF RF center distribution, histograms for horizontal and vertical positions of OFF RF centers between AD and BD. P-values represent the significance level under two-sample F-test for equal variance.

Figure4d left: As I commented above, I don't think the iGluSnFR signal has the spatial resolution of <4um (the first two points).

Re.: We disagree with Reviewer #2 on this point. Please see our responses to Comment 1&2.

Some of the figure panels may be cited out of order.

Re.: Now verified.

Reviewer #3 (Remarks to the Author):

Reviewer comments on manuscript NCOMMS-19-24867

AAV mediated transgene expression and two-photon imaging of living brain in non-human primates proved to be a huge challenge in modern neuroscience in the past decade. However, by stable GCaMP6f expression and two-photon calcium imaging of V1 in awake macaque, Tang's Lab become the first to report functional microarchitectures of thousands of neurons with single-cell resolution in behaving non-human primates (Li et al., Neuron 2017). Following this preceding seminal work, Ju et al. from the same Lab reported here that they have successfully developed a novel technique combining two-photon imaging with a glutamate-sensing fluorescent reporter, iGluSnFR, which resulting much stronger signals at neuronal dendrites when compared with classical dendritic Ca-imaging. There are three main novelties within the current study in the reviewer's opinion:

1. Ju et al. appears to be the first to apply successfully a newly developed glutamate sensor in awake behaving non-human primates.
2. By applying this newly developed powerful technique, Ju et al. mapped excitatory inputs of

different basic visual features (such as orientation, color, spatial frequency) on neuronal dendrites in superficial layers of V1 in awake behaving monkeys, demonstrating that dendritic excitatory inputs of V1 neurons encoding different basic visual features are co-existed and mutually overlapped at the circuits' level.

3. Ju et al., also found that the latency of apical inputs is temporally lagged compared to basal inputs, indicating that apical dendrites receive feedback inputs, a concept in the textbook for a long time. The evidence from Ju et al., could be the first evidence at the dendritic level from awake non-human primates by measuring the latency differences between apical and basal dendritic inputs.

Further significance evaluation:

A very recent publication in Science from Callaway's group with classical GCaMP6f expression and two-photon calcium imaging of V1 neuron soma activities in anesthesia monkeys (Garg AK, Li P, Rashid MS, Callaway EM 2019: Color and orientation are jointly coded and spatially organized in primate primary visual cortex. Science 364:1275-1279), has confirmed at the level of thousands of neurons with single-cell resolution that macaque V1 neurons can be both color and orientation tuned (e.g. Livingstone & Hubel, 1988; Leventhal et al. 1995; Johnson et al, 2001; Friedman et al., 2003; Economides et al., 2011). The nice results from Callaway's Lab suggest that orientation and color in V1 are mutually processed by overlapping circuits. The results from Ju et al. reported here in awake monkeys (the point 2 highlighted above) provide direct evidence by visualizing detailed neuronal dendritic excitatory inputs to support the long-existed concept as orientation and color in V1 are mutually processed by overlapping circuits. Therefore Ju et al. work represents a much-advanced novel approach as the first successful application of a newly developed glutamate sensor to record multiple excitatory dendritic inputs of V1 neurons in behaving non-human primates. This newly developed powerful technique for mapping excitatory dendritic inputs encoding different functional features will inevitably open a window for visualizing and probing neuronal integration at a large scale of cortical dendritic inputs' levels underlying many primate sensory and cognitive functions.

In short, I highly recommend this technically much advanced work to be published in Nature Communication for its novelties and significances. However, the authors need to do significant revision in its presentation of their work. Here are some of my major and minor comments and suggestions:

Major comments and suggestions:

Ju et al. in awake monkeys seems to be a parallel work to Garg et al., Science 2019, both using two-photo imaging to probe V1 orientation and color processing mechanisms, one at dendritic inputs level in awake monkey, the other at soma activity level in anesthesia monkeys. I derive this assumption because my impression is that the current format of Ju et al. appears to be more or less in the format of Science but wrongly submitted to Nature Communication. It must be a rejection from the previous submission to Science and resubmitted to Nature Communication with least change or revision. Also Ju et al., did not cited or even mentioned Garg et al., Science 2019 in their manuscript, suggesting that it previously submitted to Science most likely at a similar time period. My biggest concern thus is that Nothing to be shamed if being rejected by Science, instead, Ju et al. should cite and compare their results with the latest published results from Garg et al., Science 2019 as these two elegant works broadly focusing on the same topic but in complementary approaches. In fact, Ju et al. directly confirmed the assumption suggested from Garg et al., Science 2019 with a novel powerful approach in awake monkeys, suggesting a possible global computational integration mechanism that feature selectivity of neurons may result from the sum of all dendritic inputs from local and long-range neuronal circuits.

Re.: Thanks for your kind comments and suggestions. Most of our experiments were indeed performed in the second half of 2018, before the Garg et al study was published, and our manuscript was submitted before the publication of Garg et al., Science, 2019. It is indeed necessary for us to cite Garg et al.'s work in our manuscript and we have now furthermore added a detailed discussion of the two studies in context.

I would like to suggest that:

1. Combine the result section two and three into a unified section, focusing on the demonstration of the co-organized sparse and clustered dendritic inputs encoding different visual features. These results suggest that orientation, color and spatial frequency could be mutually processed at the same time when presenting combined visual stimuli.

Re.: Thanks. We agree with Reviewer #3's suggestion. However, we did the corresponding

modification in a different way. We split the original Fig. 2 into two figures, with one focusing on the interpretation of spatial clustering properties and another focusing on the functional integration between different types of inputs.

2. The sparse and clustered dendritic excitatory inputs encoding different visual cues are not necessarily competing each other, as the author claimed a trade-off mechanism. Instead, these dendritic excitatory inputs most like facilitates with each other (such as for orientation and color) when using combined visual stimuli, which exclusively happen in the natural environment (for example using colored oriented grating stimuli in a Lab setup instead, as employed by Garg et al., Science 2019). This is the actual facilitative computational mechanism at the dendritic level directly visualized by the authors in the current study.

Re.: Thanks for your constructive suggestions and sorry for the misunderstanding. We intended to express that neurons with relatively large amounts of orientation-selective inputs tended to receive homogenous orientation-selective synaptic inputs, and vice-versa for color-selective inputs. Based on the result that nearly all individual neurons received both abundant orientation- and color-selective inputs, we agree with Reviewer #3's opinion that our data could be interpreted as direct evidence for orientation/color facilitative computational mechanism at the dendritic level. We have clarified this point and modified the expression about functional trade-off in the revised manuscript.

3. They mapped the receptive field (RF) of dendritic inputs and found that the RF position varied across different dendritic input of same neuron. A previous cat area 17 study has also measured subthreshold RF inputs by in vivo intracellular recording, and they reported that size of subthreshold synaptic inputs (integration visual field) was about four times as large as the minimal discharge field (Bringuiet et al., Science 1999). Due to technical limitation, it would be impossible to map the RFs of all the input to individual, but this work gives an idea of subthreshold RF mapping on dendrites. Any possibility to get a large integration visual field based on the integration of local dendritic RFs over large cortical distance?

Re.: Thanks for this very insightful comment and suggestion. We have done this analysis and just as the reviewer predicted, we got quite a large integration field over the classical V1 neuronal receptive

fields (1.03 versus 0.34 Deg², Figure R12), thus providing a dendritic mechanism for the previously reported integration fields by Bringuier et al. (Science 1999).

Figure R12: Comparison of the local dendritic RF size versus the integration size. Each pair of dots represents data collected from one neuron.

4. Furthermore, this study compared the excitatory dendritic inputs on apical dendrites in upper layer with those on basal dendrites in the deeper layer. They found the apical dendritic inputs have similar but broader orientation preference with the basal dendritic inputs, and the apical dendritic inputs have larger RF size and lagged latency compared with basal dendritic inputs in the same neuron. It provides the first visualized evidence of temporal difference between dendrites in different depth. However, whether the recorded inputs on apical dendrites representing feedback signal and that on basal dendrite representing feedforward signal was not proved, at least not carefully discussed.

Re.: Yes, it remains challenging to directly prove input signals as feedback or feedforward ones, and our results could only bring strong but indirect evidence for this distinction. In future studies, we plan to obtain such direct evidence by combining new techniques, such as optogenetics or electrophysiology, and specific behavioral tasks, such as a top-down modulated selective attention task. We have discussed this point in greater detail in the revised manuscript.

5. This research provides a new, powerful and effective approach which can help understanding the

dendritic mechanism of signal processing on the surface of non-human primate cortex. Compared with the research of Garg et al., Science 2019 recording soma activities, this work provides much more details inside the neuronal dendrites and local circuits. However, the presentation of results (including the research background with specific scientific questions, the details of results, the description and discussion) is not clear enough. The manuscript needs major revisions. If it were presented as elegantly as Garg et al., Science 2019, the manuscript might be published already in Science.

Re.: Thanks for these constructive critiques and encouragements. We have tried our best to improve our manuscript in this revision.

Minor moments and suggestions:

1. The English writing is neither smooth nor flowing, because of the short format of the current version (it appears still in its previous Science format). Nature Communication will provide an idea platform for this work to be fully introduced, explained and discussed. The English itself must be clearer and need somebody with native English to improve it carefully and seriously.

Re.: We agree. We have taken this opportunity to present our work with a more fully developed introduction, description and discussion. In addition, native English speakers have helped us improve our grammar and the overall flow of ideas throughout the paper.

2. Which dots representing neuron I, II, III, and IV should be more clearly marked in Figure 2m, maybe also in Figure 2j and 2k.

Re.: Now clarified.

3. The figure legends should be presented more pithily. Some details of experiments could be moved into Method part, and some description of results could be moved into Result texture.

Re.: We agree. Now modified.

4. For the histogram plots in Figure 2 h & j, the scale of y axes should be 0~100, rather than 0~1 if it represents percentage (%).

Re.: Thanks, now corrected.

5. In Figure 4f (right), what is the meaning of red circle? It is not referred in the legend nor in the figure, does it represent the sampled neuron?

Re.: We apologize for the confusion. Yes, the red circle in the right panel of Fig. 4f designates the example neuron shown in the left panel of Fig. 4f. The revised figure legend now clarifies this.

6. The manuscript title could be changed to something like “From local to global: The integrated local dendritic inputs determine neural functional selectivity in primate V1”, which depicts what the authors have observed by applying their newly develop powerful technique to awake monkey V1.

Re.: We would be happy to implement this modification if the editor agrees.

Reviewers' Comments:

Reviewer #1:

Remarks to the Author:

This is beautiful work. The manuscript has been improved since the original submission and currently acceptable, I believe. I believe all my main concerns have been sufficiently addressed. Most importantly, details of the data acquisition and analyses are clearer. The new qualifier of the color vs. orientation result helps, and acceptable as is. However, I am of the opinion that they should concisely describe the main issue: Their "color selective" neurons may not be color selective, but low-pass in spatial frequency – its well-known that V1 neurons that are low-pass in spatial frequency are far more likely to be weakly tuned to orientation.

Reviewer #2:

Remarks to the Author:

All my previous concerns have been adequately addressed. I appreciate the additional analyses and the rebuttal. I have no further comments or concerns.

Reviewer #3:

Remarks to the Author:

The authors have made a significant revision, and all my questions have been fully addressed. I have no further comments and strongly support the publication of this latest two-photon imaging at the dendritic level on awake macaque V1 in Nature Communication.

Responses to Reviewers' comments:

We thank the reviewers for their kind and thoughtful critiques. All three reviewers recognized the potential significance of our contributions while bringing up several specific technical concerns. We addressed all of the reviewers' advice and revised the manuscript extensively to elucidate our findings more fully. Below we provide point-to-point responses to the reviewers' comments.

Reviewer #1 (Remarks to the Author): This paper measured the spatial and functional distribution of excitatory inputs to neurons in superficial V1. Applying this technique to the primate is extremely impressive and allows for one to address how tuning arises in the most relevant animal model to human visual perception. There is sufficient data from this challenging technique. My first concern is that they did not give enough attention to signal extraction from their images. The off line processing is not a trivial endeavor, but I feel is necessary to make sense of this impressive data set. Additionally, I have major concerns about their color stimulus paradigm and the conclusions that can be drawn with respect to the segregation of form and color selectivity. I think this paper would be of better service if more attention was instead paid to a rigorous signal extraction from the movies.

Re.: We thank reviewer #1 for commending the significance and novelty of our work. We have now enhanced the rigor of the signal extraction analyses from the raw data. We have now also provided further description of how we segregated form vs color selectivity tuning from the data. Details below.

Major concerns:

1) Signal extraction

My main question is whether the ROI signals are independent from each other and the background. It seems challenging to tease these apart since the ROIs are very small - even the smallest of smearing artifacts (such as brain motion in the Z-axis) is going to mix the signals. Furthermore, the continuity of the dendritic tree in each (averaged) image implies either a wide depth-of-field and/or Z motion. Although perhaps challenging, a more rigorous method of signal processing is warranted for this valuable data set. I have attempted to provide examples of how I think this could be improved, but

the authors could certainly have better ideas.

Re.: In our experiments, individual inputs showed significant tuning to orientation, spatial frequency or color, as shown in Figure 1C-D. Given that the stimuli were presented randomly, and taking into account the high repeatability of the single-trial fluorescence traces (Supplementary Fig. 2), our data indicate that the signals we recorded did not arise from brain motion related changes, which we rule out given that such artifacts cannot explain the selectivity to visual stimulus features found in our data.

We do agree that spatial resolution is important for high quality sampling of spine data, and we here show that our imaging resolution is sufficiently high to tease apart different ROIs. Our two-photon microscope (Bruker Nano, Inc., Prairie Ultima IV two-photon microscope) has been calibrated to capture high resolution two-photon images: The point-spread function (PSF) has been measured at: $0.4\mu\text{m}$ X-Y plane resolution, and $2.0\mu\text{m}$ Z axis, when using the 25x Olympus objective used for these experiments. This is a typical range of measurements (compared to other studies in the field), and we have achieved the same diffraction-limited microscopic resolution—near the physical limit also found in other similar two-photon systems—reported in prior work employing the imaging of dendrites, spines, and other filaments and small cells in the neuropil. In our dendritic imaging experiments, the sampling resolution was set to about 4 pixels/ μm . This setting oversampled to magnify beyond the physical limitation of the two-photon imaging system, which is a standard approach. With this approach, single-ROI signals were robust and repeatable (Figure 1C, 1E and S2). We also replicated these results using a higher power objective combined with higher magnification to achieve higher spatial resolution (18 pixels/ μm), using a $60\times$ objective lens under zoom $8\times$ and 512×512 pixels (Figure R1). Because those recordings produced similar results, we conclude that the results with the 25X objective have the necessary precision for spine analysis, and thus represent accurate and valid results.

Figure R1. Left, an averaged image obtained with a 60 \times objective lens under zoom 8 \times , covering a field of view of 28 μm , averaged across 1000 frames, recorded at a depth of 192 μm . Right, the differential image (ON-OFF) under one visual stimulus exhibited hotspots on target dendrites.

As for the concern about potential z-axis fluorescence leaks across different depths, we can rule out the possibility that our data is systematically contaminated by out-of-plane recording artifacts because of the fine optical sectioning obtained in two-photon imaging (a function of the PSF along Z axis $\sim 2.0\mu\text{m}$ for our system). Figure R2 shows a series of continuous Z-stack images, with each pair of neighboring images having a depth difference of $3\mu\text{m}$: these images vary greatly and thus are not mixing beyond the z-axis diffraction-limited resolution of our 25X objective. This control experiment supplies a direct impression of fine imaging resolution along the Z axis. To obtain additional dendritic inputs in one experimental session, we always chose a target focal plane where longer continuous dendritic trees existed (as an example, we set the imaging depth at $127\mu\text{m}$ for the neuron in Figure R2). Hence, the continuity of the dendritic tree arises from the intrinsic neuronal morphology and from the imaging plane we chose, rather than from a wide depth-of-field or from Z motion. As with all two-photon recordings, fluorescence and signal crosstalk may have occasionally occurred across different depths, but the likelihood that this caused a regular or systematic contamination of our results is very low, and it would mean that there were hidden invisible dendrites aligned with our dendrites, that we could not see in our Z-stacks. We do not believe that this could have occurred in any systematic fashion that we could miss, especially as it would mean that these hidden dendrites nevertheless are somehow contributing a fluorescence artifact significant enough to influence our results. Therefore, such a possibility is vanishingly low given the outcome of our controls. As also shown in Supplementary Fig. 2 c-g, the background fluorescence change was minute compared to the robust signals we found on our dendritic recordings. Additionally, regions

that exhibited robust fluorescence in our analyses arose in positions matching our dendritic branches (Figure R1, Supplementary Fig. 2 b-c, Video 1 and 2), arguing against the possibility that our signals originated in planes that were out of focus.

Figure R2: Averaged images obtained with a 25x objective lens under zoom 1x, covering a field of view of 128 μ m, averaged across 100 frames each, recorded at a 3 μ m depth interval from 120 μ m to 132 μ m.

We nevertheless agree that our study would greatly benefit from more rigorous signal processing, and we have incorporated such methods and analyses in our revised MS (please see the corresponding subheading below).

A) Can you apply a more rigorous method to extract independent signals on the dendrite, such as ICA? Perhaps they could apply the pipeline in this paper:

Pnevmatikakis et al. Simultaneous Denoising, Deconvolution, and Demixing of Calcium imaging data.

Re.: Following your kind suggestion, we have now employed Pnevmatikakis et al's extraction procedure for each individual neuron, and re-processed all corresponding statistics as a function of the new ROIs, modifying the relevant figures accordingly in the revised manuscript (Fig. R3). The results continue to support all our previous conclusions, with the exception of the significance level in the previous revision's Fig. 4d, which fell from from $p=0.047$ to $p=0.18$ (Figure R4). We assume this occurred because we changed the method of characterizing ROIs, and thus the ROI population changed; because we continued to exclude those neurons with fewer than 15 ROIs on their Apical Dendrites (ADs), the sampling size of analyzed neurons fell from 9 neurons to 6. Though the tendency of biased distribution still existed, the power value for this t -test equaled 0.21 (with the hypothesis that AD beta value differed significantly from Basal Dendrites (BDs)), indicating a lack of

sampled neurons. We therefore removed this panel (original Fig. 4d) from the manuscript and we will readdress this specific analysis in the future, when there is sufficient power to conduct a proper analysis using Pnevmatikakis et al's method.

Figure R3: ROIs with Pnevmatikakis et al's method to extract independent signals on the dendrites.

Figure R4: The removed panel from original Fig. 4d. Left, relationship between the dendritic distance of ROI pairs and their orientation tuning correlation coefficients on AD (red) and BD (blue) of the example neuron in Fig. 4. β -value represents the correlation baseline (see Methods). Error bar, SEM. Right, AD β -value versus BD's of 6 neurons (neurons with less than 15 ROIs on their AD were discarded). The example neuron is denoted with a red circle.

B) As it stands, the background signal ("local map") is very mysterious. I would like, at the very least, to have a better understanding of it. If it is very weak, it seems that the local maps could mostly just be a heavily smeared version of the "output neuron". On the other hand, if it is very strong, that creates a greater concern that it is bleeding into their ROIs.

How weak or strong is this background signal? Perhaps they could start by just showing an image of

SNR... do the dendrites “pop out”. Also, what does the orientation map look like without the smoothing?... is it mostly just noise outside of the ROIs? Showing this for every neuron would be most useful. Also, perhaps they could, in words, speculate on the physiological origin of this background signal.

Re.: As to the concern about the local orientation maps, the first point we should clarify is that orientation map measurements and dendritic imaging were conducted in separate experiment sessions with distinct apparatus settings, as required by the differing resolutions and imaging conditions for the different kinds of data. Specifically, high resolution dendritic imaging was conducted using a 25× objective lens (1.1 N.A., Olympus) with zoom 4× scanning mode, covering a field of view of 136μm × 136μm. Because a hyper-column in macaque V1 takes about 1mm² of cortical area, we obtained maps of orientation pinwheel structure using a much larger field of view (about 6.25X larger): a 16× objective lens (0.8-N.A., Nikon) with zoom 1×, which covered a field of view of 850μm × 850μm (left panels in Figure R5). Smoothed orientation maps of these recording sites show clear orientation pinwheel structure (right panels in Figure R5). In addition, as the reviewer requests, here we draw pixel-by-pixel orientation maps to give a first glance of the raw orientation preference without smoothing (middle panels in Figure R5), which also demonstrate a structured spatial clustering in consistence with the smoothed patterns. The recording depths for each imaging site were all set to around 200μm. Then, each neuronal position was aligned to a corresponding imaging plane along the Z-axis, and local maps for each target neuron were determined. Since individual dendritic branches were relatively tiny at this imaging scale level, local orientation maps were measured from global membrane-located iGluSnFR fluorescence.

Figure R5: Orientation maps for each imaging site from 3 monkeys, recorded with a 16 \times objective lens (0.8-N.A., Nikon) under zoom 1 \times , covering a field of view of 850 $\mu\text{m} \times 850\mu\text{m}$. For each imaging site, left panel: an averaged image from all frames acquired during corresponding recording session; middle: pixel-by-pixel orientation map without smoothing; right: orientation map after Gaussian filtering, black circles represent the cortical position of each target neuron.

As for the concern about relative strength of background signals versus dendritic signals during high resolution dendritic imaging, we found that signals on dendritic branches were much stronger and could be isolated reliably (Supplementary Fig. 2 a-g, see also Video 1 & 2). As the reviewer suggested, we also calculated SNR maps for individual neurons, and signals on dendrites indeed popped out (Figure R6). It is worth noting that a few hot-spots outside target dendritic regions exhibited strong activity during visual stimulation, though such active regions could be reliably excluded from targeted dendritic inputs for each single neuron (Figure R1 A-F), based on their spatial locations.

Figure R6: SNR maps of iGluSnFR fluorescence signals. **a** Left, two-photon image of an example neuron at a depth of 127 μm , averaged across 1000 frames from one recording session. Right, pixel-by-pixel SNR map of fluorescence signals under the same field of view. For each pixel on this map, an orientation SNR index was defined as the ratio of fluorescence variance between averaged ON and OFF frames of 12 orientations under the preferred spatial frequency, and a color SNR index was determined as the ratio of fluorescence variance between averaged ON and OFF frames of 9 colors. The SNR value of each pixel was then set as the maximum between the corresponding orientation SNR index and color SNR index. SNR value is denoted by the colorbar on the right side. **b** Same as **a**, for another sample neuron at a depth of 105 μm .

2) Color vs. orientation dichotomy

I have issues with the interpretation of one of their main results. The conclusion from Fig. 2 is a dichotomy between the different inputs: inputs are either selective for orientation or color. I don't think its possible to make this claim without colored gratings. To summarize, I think its quite possible that they are instead (or in addition) looking at a difference in low-pass vs. band pass spatial frequency tuning - some are tuned for low pass (full-field), whereas others are tuned to bandpass (gratings). If a cell is low-pass for spatial frequency, it will have weak orientation selectivity, but still respond to one of the colors. For example, the orientation selective inputs may be selective to one of their colors, but the full-field stimuli elicit almost no response. In addition, some of the color selective inputs could be orientation selective, but not to monochromatic gratings.

Re.: Thank you for pointing out that it is less rigorous for us to describe inputs as either orientation- or color-selective without using colored gratings. We have now re-analyzed our data using Pnevmatikakis et al.'s procedure, and we have also introduced an orientation versus color index to better illustrate the functional preference of individual synaptic inputs. We found that the preference between orientation versus color was quantified continuously, and we could avoid creating a false dichotomy unintentionally (Fig. 3b). Furthermore, we should point out that previous studies have found color-orientation feature integration only in Layer 2/3 neurons (Leventhal et al., J

Neurosci, 1995; Friedman et al., J Physiol, 2003; Johnson et al., J Neurosci, 2008), including a recent study that used cellular-resolution two-photon imaging to show that orientation and color are jointly coded by V1 superficial neurons (Garg et al., Science, 2019). Our results are thus consistent with these studies and moreover present direct evidence of dendritic-level computations for orientation versus color processing mechanisms in V1 neurons. Also, following from the updated analysis, a large proportion of inputs tuned to both orientation and color, which makes sense given that the previous research suggests that these inputs most likely come from Layer 2/3.

We agree that we may have potentially underestimated the proportion of inputs selective to both orientation and color in our previous revision. We are also grateful that Reviewer #1 suggested we might instead look at a difference between low-pass versus band-pass spatial frequency tuning. However, without colored grating stimuli, it is also not enough to assign an input as low-pass or band-pass selective, as color and SF are conflated (as the reviewer correctly pointed out). For example, a color-selective input may respond to both full-field color patches and colored gratings but not to monochrome gratings, which would make it both low-pass and band-pass selective, while it would be determined solely as low-pass selective without colored grating stimuli. Considering that orientation-color co-tuned inputs could be regarded as both orientation-selective and color-selective, our main conclusion that nearly all individual neurons received both abundant orientation and color inputs remains valid. Even so, we have now modified our description in the manuscript to avoid misinterpretation.

3) Other questions about the color paradigm

-How was color selectivity computed?

Re.: Thank you for pointing out that we missed a detailed description about color selectivity. To determine the color selectivity of each synaptic input, a selectivity index was calculated as $1 - (\text{minimum response})/(\text{maximum response})$ (Tomoyuki et al., J Neurosci, 2014). Also, a signal-to-noise-ratio (SNR) index of single synaptic inputs was defined as the ratio of fluorescence variance between averaged ON and OFF frames of all colors. Moreover, we conducted a balanced one-way analysis of variance (ANOVA) on single trial response values to obtain inputs with significant differences in response strength across various colors. When the color selectivity index was larger than 0.8, SNR was higher than 1 and response variation across colors was significant

(ANOVA, $p < 0.05$), we assigned these synaptic inputs as color-selective. This is now clarified in the revised manuscript.

-More detail in the methods is warranted regarding how each color was generated on the monitor. e.g. what was the background luminance? Presumably it wasn't 8cd/m².

Re.: Agreed, a more detailed description of color generation is required. The background luminance was set to 25 cd/m² for all our experiments on a 17-inch LCD monitor (Acer V173, 80Hz refresh rate and 1280 × 960-pixel resolution). Full saturated colors were generated by setting the values of the RGB channel to [255, 0, 0] - red, [255, 127, 0] - orange, [255, 255, 0] - yellow, [0, 255, 0] - green, [0, 255, 255] - cyan, [0, 0, 255] - blue, [255, 0, 255] - purple, [255, 255, 255] - white and [0, 0, 0] - black. We then measured the chromaticity and luminance of each saturated color using SpectroCAL MKII Spectroradiometer (Cambridge Research System). Accordingly, we set all equiluminant colors to match the blue channel, which had the weakest luminance value at 8.2 cd/m². To calibrate the other equiluminant colors, we adjusted the RGB channel values for each color to a luminance intensity as 8.2 cd/m², while maintaining chromatic saturation of each color channel. We have now added these methods to the revised manuscript.

-Its hard to believe that they got the same results for the saturated and unsaturated case since the contrasts will be dramatically different. They should elaborate on what they mean by “did not vary significantly”. Does the presented data come from one or the other, or some combination?

Re.: Thank you for pointing out that this conclusion requires further data to support it. We now present our analysis in the revised manuscript (and details below). The data presented in the main figures came from saturated color. At the beginning of our experiment (on Monkey 1 and Monkey 2), we only used saturated colors. However, later we realized that it was necessary to eliminate the effect of luminance in our experiment. We then did control experiments to compare the difference between saturated vs. equiluminant color-evoked dendritic responses (on Monkey 3). Color preference of 79% dendritic inputs stayed stable when comparing saturated vs. equiluminant stimuli (see below in Figure R7b, bottom panel, for error tolerance between neighboring colors, red shadow). Moreover, the averaged response intensity of dendritic inputs did not vary significantly under saturated versus

equiluminant chromatic stimuli ($p > 0.05$ by Wilcoxon rank sum test, Figure R7c, e). This result makes intuitive sense given the fact that, even in our equiluminance condition, which is less bright than the saturated condition, the luminance was nevertheless set at a relatively high luminance because it is the highest luminance setting of the blue channel, which is perceptually quite bright. Thus, we should not expect a significant difference between the saturated and equiluminant conditions.

Figure R7: Comparison between non-equiluminant and equiluminant chromatic stimulus evoked responses. a Peak response intensities of ROIs to equiluminant color stimuli versus non-equiluminant (but colored) stimuli. Red line, linear regression. 7 neurons in total from Monkey 3 were recorded. Bottom, statistics for all ROIs from all 7 neurons. b Color preference of each ROI among equal-luminance colors versus non-equal-luminance colors. Brightness denotes ROI proportion, referring to colorbar above the bottom figure. c Averaged response intensity of all ROIs under non-equiluminant versus equiluminant chromatic stimuli. ROIs in a-c were collected only according to non-equiluminant color stimuli. Asterisk, significant difference, by Wilcoxon rank sum test; $p=0.11$ when combining all ROIs. d Non-equiluminant and equiluminant chromatic stimuli response strengths within ROIs that were tested with both. e, Same as c, statistics of averaged response intensities; $p=0.07$ when combining all ROIs.

4) Interpretation of a hierarchy

The authors interpret the broad, yet independent functionality of inputs in terms of a hierarchy. They are recording from the upper margin of layer 2/3, which receives input from the bottom of 2/3, not 4C. It seems odd that functional segregation would remain so rigid until this stage. Instead, as described in '2' above, it may not be a segregation of color vs orientation selective inputs, but instead low-pass vs. bandpass inputs which are two sides of a continuum.

Re.: We agree with Reviewer #1's comment about a functional hierarchy and that we did not explain it carefully in the original manuscript. Since we have mapped ON/OFF receptive fields for each input, we tried the methods used in 'Kuo-Sheng Lee et al., Nature, 2016' to classify inputs as simple or complex (namely inputs from simple or complex cells), and found that nearly all inputs are complex inputs. Taking into account that neurons in layer 4 are mainly simple cells, and that complex cells appear on layer 2/3, we agree that most of the dendritic inputs we recorded originate from the bottom of layer 2/3 rather than from layer 4C. We also agree that, without using colored grating stimuli, our conclusion of a functional segregation of orientation versus color selective inputs lacked rigor. We have now modified the manuscript to address this limitation. We also want to thank the reviewer for voicing this criticism, as we are now considering a detailed follow-up study about orientation and color segregation using colored grating stimuli.

Reviewer #2 (Remarks to the Author):

Manuscript by Ju et al., titled “Spatiotemporal functional organization of excitatory synaptic inputs onto macaque V1 neurons” describes spatial distribution patterns of functionally characterized synaptic inputs over the dendritic arbors of V1 superficial layer neurons in Macaques. The authors performed in vivo dendritic input mapping, using two-photon imaging of recently developed genetically-encoded glutamate sensor, iGluSnFR, while the awake monkeys viewed visual stimuli with various features. They report that the majority of the visually-responsive hot spots on the dendrites was either orientation-selective or color-selective and rarely both, and that there was local clustering of hot spots responding to similar stimulus feature. Moreover, differential response properties such as receptive field size and response latencies were observed between apical vs basal dendrites, indicative of differential roles for these distinct groups of dendrites in sensory integration. As far as I am aware, this is the first functional input mapping study performed on a primate with neural activity indicators of any kind, and would be of interest to a broad audience in neuroscience community.

However, several key points remain questionable that need to be addressed, including the type of neurons that were being imaged. The nature of glutamate signals that were measured also needs to be clarified and addressed as some of the known kinetics of the indicator may render it unsuitable for precise input mapping in vivo let alone at the resolution scale used by the authors. Specific comments both major and minor are as listed below:

Major comments:

1) The identities of the neurons imaged in this study:

The type of cells imaged in the study is only vaguely described as “superficial layer neurons” and, as the authors state in the text, includes both neural types with spiny and smooth dendrites. This is problematic as the former is likely excitatory neurons while the other could be inhibitory neurons. These two opposing neuronal types should not be pooled together when investigating the functional organization of synaptic inputs over a dendritic arbor as they are known to have very different response properties to begin with. The authors report analyzing a total of 23 neurons from 3 different monkeys? Of these, how many neurons were putative excitatory (spiny) vs inhibitory (smooth) neurons? It would’ve been insightful if post hoc immunohistochemistry was performed to determine the identities of these neurons, but at the very least, the data should be separated into these two groups before analyzing features such as local clustering of preferred stimulus features.

Interneurons in other species have been shown to have longer stretches of the dendritic shafts responding to similar stimulus features (10-20um) while the overall response properties may be heterogenous (Chen et al., Nature 2013) which could no doubt influence the synaptic clustering analyses as shown in Figure 2f.

Re.: Thanks for your suggestion! We agree that it is quite important to identify the neuronal type and conduct data analyses separately for different types of neurons. Reviewer #2's specific concern was the previously reported homogeneous orientation tuning along continuous domains on inhibitory neuron dendrites (Chen et al., Nature, 2013), and how this might influence the synaptic clustering analyses. It is worth noting that, whereas Chen et al. identified signals on spines of excitatory neurons as synaptic inputs, they did not identify equivalent signals from large dendritic domains. Instead, they identified dendritic-spike induced fluorescence changes. This is clarified in the last paragraph of 'Calcium transients in GABAergic dendrites' section in Chen et al., in which they state that the large size of their domains might reflect spatial clustered homogeneous inputs or a few sparse, randomly distributed strong inputs, amplified by local postsynaptic mechanisms. Hence, from Chen et al, the spatial organization of inputs onto inhibitory neurons remains unknown, and the inference that there would be significant difference of spatial clustering properties between inputs onto excitatory versus inhibitory neurons is not clear. We also disagree with this interpretation because Ca-imaging experimenters post-process their data to control for the effects of back-propagating spikes, and this is a large source of noise (see Fig. 5d in Chen's paper for an example, which suggests that even on dendritic spines, the input signals they could obtain were pretty sparse). By contrast, iGluSnFR resulted in strong signals on neuronal dendrites and maintained single-input measuring resolution in the meantime, which tends to be a more reliable sensor for dendritic input imaging. Thus, though we have no quarrel with the Chen et al. data, it is not necessarily true that their results differ from ours, due to the discrepancy in techniques used in each study.

Having said this, we have now done the analysis as Reviewer #2 requested. We classified neurons as either smooth or spiny ones according to their dendritic spine densities (11 neurons were determined as smooth ones and 12 as spiny ones, Figure R8a). Then, by comparing the λ -value (spatial length constant) and β -value (correlation baseline) of spatial tuning correlation between smooth versus spiny neuronal classes (Figure R8 b), no significant difference was observed for

orientation tuning spatial clustering constant between smooth and spiny neurons (λ -value, Figure R8b).

Currently, it remains a major technical challenge to target specific types of neurons in macaque cortex *in vivo* (Dimidschstein *et al.*, *Nat. Neurosci.*, 2016; Nagai *et al.*, *Biochemical and Biophysical Research Communications*, 2019). Due to the technical limitations of genetic selection within macaque cortical neurons, nearly all two-photon imaging studies in this field could only make general conclusions across all types of neurons (Nauhaus *et al.*, *Nat. Neurosci.*, 2012; Sadakane *et al.*, *Cell Report*, 2015; Seidemann *et al.*, *eLife*, 2016; Garg *et al.*, *Science*, 2019). This remains true for our study, and the revised manuscript now provides a more detailed discussion of these factors.

Figure R8: Comparison of functional organization between synaptic inputs onto smooth versus spiny dendrites. a, Spine densities of the recorded neurons. Neurons with dendritic spine density lower than 0.3 units per 10 μm were assigned as smooth ones (putative inhibitory, marked in blue), and those with spine density higher than 0.3 as spiny ones (putative excitatory, marked in red). b, Left, relationship between the dendritic distance of inputs and their orientation tuning correlation coefficients within smooth neurons (top, in blue) and spiny neurons (bottom, in red). Right, statistics (mean value and 95% confidence interval) of λ -value and β -value of dendritic orientation tuning for each kind of neurons.

2) The use of iGluSnFR and dendritic shaft imaging for input mapping:

The authors used a glutamate sensor, iGluSnFR, and all the measurements were made from the “hot spots” on the dendritic shafts rather than the dendritic spines themselves.

While this may be less than ideal but acceptable approach for looking at more general organizational patterns of synaptic inputs, this particular reporter is not suitable for investigating local clustering of inputs due to its high affinity to glutamate which makes it even sensitive to the small changes in the

glutamate concentration outside of the synaptic cleft like glutamate spillover. The reported $1/e^2$ decay distance for GluSnFR is $\sim 3.6\mu\text{m}$ (Kazemipour et al., Nature Methods 2019). The clustering of dendritic regions responding to similar stimulus features reported in this study is well within this distance, meaning that the reporter is unlikely to have the resolving power to support their findings. This, combined with the point made under Comment 1, leaves the authors conclusions regarding the synaptic clustering weak and unsupported.

Re.: Please note that, while the $1/e^2$ decay distance for GluSnFRs was $3.6\mu\text{m}$ as illustrated by Kazemipour et al. (corresponding cross-validation tuning correlation value higher than 0.06), the half-life ($1/2$) decay distance is less than $1\mu\text{m}$ (correlation value higher than 0.22). In other words, whereas iGluSnFR could perceive small amount of glutamate that spread up to $3.6\mu\text{m}$ away, robust detection of glutamate occurred primarily within $1\mu\text{m}$. This is supported by Kazemipour et al.'s work as our results are consistent and we obtained similar sizes for inputs onto neuronal dendrites from our iGluSnFR imaging experiments (the mean value of ROI radius equaled $0.76\mu\text{m}$, Figure R9). Such spatial scale of iGluSnFR signals could then support our clustering analysis. Combined with our reply to Reviewer #2's concern under Comment 1, we conclude that it is indeed reasonable to analyze the spatial clustering in our manuscript given our analyses of Reviewer #2's concerns, which we have now discussed thoroughly in the revised manuscript.

As for the concern about input signal positions on dendrites, we wish to clarify that both our data and Kazemipour et al.'s contain inputs onto both dendritic shafts and spines (see ROI 1 in Fig. 1, see also Fig. 2 b-e and Supplementary Fig. 6a; Fig. 4c in Kazemipour et al.'s work). Compared to traditional Ca-imaging, imaging with glutamate sensor iGluSnFR does have a heightened sensitivity to glutamate inputs as iGluSnFR eliminates the need to post-process the data and control for back-propagating spikes (see Supplementary Fig. 6, also see the visible difference shown in Fig. 4c in Kazemipour et al., Nature Methods, 2019). For dendritic input imaging, it is an advantage rather than a disadvantage to obtain signals on dendritic shafts, since if there are glutamate inputs on shafts, they are excitatory inputs that should be accounted for.

Figure R9: Size statistics for all 1818 ROIs obtained from 3 monkeys. ROI radius was defined as the square root of corresponding ROI area. The mean value of ROI radius was 0.76 μm, the minimum was 0.53 μm and the maximum was 1.17 μm.

3) Regarding functional trade-off between orientation/color inputs and the specificity of the glutamate signals: The reported trade-off between the two stimulus features is intriguing, but I am having difficulty imagining how this could be especially on apical dendrites which supposedly receive lateral inputs from other cortical neurons such as another superficial layer neurons that are tuned to both color and orientation. Can the authors provide plausible explanations for how this could be?

Re.: Thank you for the opportunity to clarify this issue. The conclusion about orientation/color functional trade-off was found only on inputs to basal dendrites, not apical dendrites. It was not until Fig. 4 that inputs onto apical dendrites were analyzed. We have now clarified this point in the revised version.

Rather, given the observation reported in this study that “neurons in iso-domain of orientation column....tended to receive more oriented inputs” rather than color inputs, is it possible that the iGluSnFR is sensing not only glutamate release from those inputs synapsing onto the target neuron but also the spillovers from other axons in the vicinity synapsing onto non-target (unlabeled) nearby neurons (meaning not all the signals detected on the imaged neuron are bona fide synaptic inputs)? Glutamate sensing is tricky because the sensor is membrane bound and reports the level of glutamate OUTSIDE the cell and the more sensitive the reporter is, the more likely it is to pick up non-synaptic glutamate which would be abundant in vivo, especially in iso-domain part of the orientation column stimulated by the preferred orientation stimulus.

Re.: As indicated in our reply to Comment 2, the highly localized iGluSnFR signals (with a mean value of ROI radius equaling 0.76 μm , Figure R9) provide strong evidence for local glutamate release, rather than spillover from nearby synaptic clefts. Representative movies of the iGluSnFR recordings for iso-domain neurons directly show how local iGluSnFR signals integrate in the dendrite (Supplementary Video. 2). Even the strongest signals did not spillover to nearby locations. Because clearly identified inputs do not spillover to other clearly identified nearby locations, even during strong stimulation, it follows that our other recordings are not driven by misattributed spillover.

Minor comments:

Line 48: “The temporal sequencing of dendritic inputs will lead to new and critical insights” – This statement needs to be clarified as I am not sure what is meant by it.

Re.: We agree. Now rectified in the revised manuscript.

Line 78-81: “...our data showed that summed input orientation preferences matched the orientation preference of the integrate somal Ca response...was moderately broader than somatic orientation selectivity.” – Was is the case for both neurons residing within the iso-domain and for those in the pinwheel centers?

Re.: Yes. In Monkey 3, we collected 6 neurons that co-expressed RCaMP and iGluSnFR, and 5 neurons among them were orientation selective (same evaluation criteria as the one we used to classify dendritic inputs). For these neurons, somatic orientation preference matches well with the summed input orientation preference (Figure R10a), and the summed input orientation selectivity was always broader than somatic orientation selectivity regardless of the neuron's relative position in the orientation column structure (Figure R10b).

Figure R10: Functional comparison between RCaMP and iGluSnFR signals. a, Somatic orientation preference versus that of summed dendritic ROI responses for each single neuron. b, Full width at half maximum (FWHM) of somatic orientation tuning (in red) or summed dendritic ROI orientation tuning (in green) versus neural distance to pinwheel center. Black dashed line links data dots from same neuron.

Line 191: “...one local pair of synaptic inputs might share similar orientation preferences while possessing quite dissimilar RFs” – Previous studies have discovered OFF receptive field-anchoring in which neurons in an orientation column tend to have OFF subregions that are aligned in visual space and ON subregions that are displaced. If you analyzed ON and OFF subregions separately rather than simply using averaged RF centers (Fig 3d, e), do any additional patterns emerge?

Re.: The reviewer may refer to the study done by Lee et al., Nature, 2016. We have also tried to analyze ON and OFF subregions separately, however, we found no significant difference between them, possibly due to the fact that most of the synaptic inputs we recorded were classified as complex inputs (using the same classification pipeline as Lee et al. described in their Methods), whereas Lee et al.'s conclusion was based on simple cells. Therefore, our two studies may be examining different levels in the visual processing hierarchy.

Figure4: If you analyzed ON and OFF subregions separately, do you see any differential spatial distribution patterns of each subregions between apical vs basal dendrites?

Re.: We have done such analyses (Figure R11), and found the distribution of ON and OFF RFs more scattered for inputs onto apical dendrites only in 3 neurons out of 9 that we collected in total ($p < 0.05$,

two-sample F-test for equal variances). Our study did not set out to test the Lee et al's hypothesis and it thus does not have the power to make a robust conclusion on the basis of the present data.

Two-sample F-test for equal variances

Figure R11: Spatial distribution of ON and OFF subregions for inputs onto apical (AD) versus basal dendrites (BD). From left to right, these panels represent for: ON/OFF mean RF center distribution (shown as eccentricity in degree), ON RF center distribution, histograms for horizontal (x) and vertical (y) positions of ON RF centers between AD and BD, OFF RF center distribution, histograms for horizontal and vertical positions of OFF RF centers between AD and BD. P-values represent the significance level under two-sample F-test for equal variance.

Figure4d left: As I commented above, I don't think the iGluSnFR signal has the spatial resolution of <4um (the first two points).

Re.: We disagree with Reviewer #2 on this point. Please see our responses to Comment 1&2.

Some of the figure panels may be cited out of order.

Re.: Now verified.

Reviewer #3 (Remarks to the Author):

Reviewer comments on manuscript NCOMMS-19-24867

AAV mediated transgene expression and two-photon imaging of living brain in non-human primates proved to be a huge challenge in modern neuroscience in the past decade. However, by stable GCaMP6f expression and two-photon calcium imaging of V1 in awake macaque, Tang's Lab become the first to report functional microarchitectures of thousands of neurons with single-cell resolution in behaving non-human primates (Li et al., Neuron 2017). Following this preceding seminal work, Ju et al. from the same Lab reported here that they have successfully developed a novel technique combining two-photon imaging with a glutamate-sensing fluorescent reporter, iGluSnFR, which resulting much stronger signals at neuronal dendrites when compared with classical dendritic Ca-imaging. There are three main novelties within the current study in the reviewer's opinion:

1. Ju et al. appears to be the first to apply successfully a newly developed glutamate sensor in awake behaving non-human primates.
2. By applying this newly developed powerful technique, Ju et al. mapped excitatory inputs of

different basic visual features (such as orientation, color, spatial frequency) on neuronal dendrites in superficial layers of V1 in awake behaving monkeys, demonstrating that dendritic excitatory inputs of V1 neurons encoding different basic visual features are co-existed and mutually overlapped at the circuits' level.

3. Ju et al., also found that the latency of apical inputs is temporally lagged compared to basal inputs, indicating that apical dendrites receive feedback inputs, a concept in the textbook for a long time. The evidence from Ju et al., could be the first evidence at the dendritic level from awake non-human primates by measuring the latency differences between apical and basal dendritic inputs.

Further significance evaluation:

A very recent publication in Science from Callaway's group with classical GCaMP6f expression and two-photon calcium imaging of V1 neuron soma activities in anesthesia monkeys (Garg AK, Li P, Rashid MS, Callaway EM 2019: Color and orientation are jointly coded and spatially organized in primate primary visual cortex. Science 364:1275-1279), has confirmed at the level of thousands of neurons with single-cell resolution that macaque V1 neurons can be both color and orientation tuned (e.g. Livingstone & Hubel, 1988; Leventhal et al. 1995; Johnson et al, 2001; Friedman et al., 2003; Economides et al., 2011). The nice results from Callaway's Lab suggest that orientation and color in V1 are mutually processed by overlapping circuits. The results from Ju et al. reported here in awake monkeys (the point 2 highlighted above) provide direct evidence by visualizing detailed neuronal dendritic excitatory inputs to support the long-existed concept as orientation and color in V1 are mutually processed by overlapping circuits. Therefore Ju et al. work represents a much-advanced novel approach as the first successful application of a newly developed glutamate sensor to record multiple excitatory dendritic inputs of V1 neurons in behaving non-human primates. This newly developed powerful technique for mapping excitatory dendritic inputs encoding different functional features will inevitably open a window for visualizing and probing neuronal integration at a large scale of cortical dendritic inputs' levels underlying many primate sensory and cognitive functions.

In short, I highly recommend this technically much advanced work to be published in Nature Communication for its novelties and significances. However, the authors need to do significant revision in its presentation of their work. Here are some of my major and minor comments and suggestions:

Major comments and suggestions:

Ju et al. in awake monkeys seems to be a parallel work to Garg et al., Science 2019, both using two-photo imaging to probe V1 orientation and color processing mechanisms, one at dendritic inputs level in awake monkey, the other at soma activity level in anesthesia monkeys. I derive this assumption because my impression is that the current format of Ju et al. appears to be more or less in the format of Science but wrongly submitted to Nature Communication. It must be a rejection from the previous submission to Science and resubmitted to Nature Communication with least change or revision. Also Ju et al., did not cited or even mentioned Garg et al., Science 2019 in their manuscript, suggesting that it previously submitted to Science most likely at a similar time period. My biggest concern thus is that Nothing to be shamed if being rejected by Science, instead, Ju et al. should cite and compare their results with the latest published results from Garg et al., Science 2019 as these two elegant works broadly focusing on the same topic but in complementary approaches. In fact, Ju et al. directly confirmed the assumption suggested from Garg et al., Science 2019 with a novel powerful approach in awake monkeys, suggesting a possible global computational integration mechanism that feature selectivity of neurons may result from the sum of all dendritic inputs from local and long-range neuronal circuits.

Re.: Thanks for your kind comments and suggestions. Most of our experiments were indeed performed in the second half of 2018, before the Garg et al study was published, and our manuscript was submitted before the publication of Garg et al., Science, 2019. It is indeed necessary for us to cite Garg et al.'s work in our manuscript and we have now furthermore added a detailed discussion of the two studies in context.

I would like to suggest that:

1. Combine the result section two and three into a unified section, focusing on the demonstration of the co-organized sparse and clustered dendritic inputs encoding different visual features. These results suggest that orientation, color and spatial frequency could be mutually processed at the same time when presenting combined visual stimuli.

Re.: Thanks. We agree with Reviewer #3's suggestion. However, we did the corresponding

modification in a different way. We split the original Fig. 2 into two figures, with one focusing on the interpretation of spatial clustering properties and another focusing on the functional integration between different types of inputs.

2. The sparse and clustered dendritic excitatory inputs encoding different visual cues are not necessarily competing each other, as the author claimed a trade-off mechanism. Instead, these dendritic excitatory inputs most like facilitates with each other (such as for orientation and color) when using combined visual stimuli, which exclusively happen in the natural environment (for example using colored oriented grating stimuli in a Lab setup instead, as employed by Garg et al., Science 2019). This is the actual facilitative computational mechanism at the dendritic level directly visualized by the authors in the current study.

Re.: Thanks for your constructive suggestions and sorry for the misunderstanding. We intended to express that neurons with relatively large amounts of orientation-selective inputs tended to receive homogenous orientation-selective synaptic inputs, and vice-versa for color-selective inputs. Based on the result that nearly all individual neurons received both abundant orientation- and color-selective inputs, we agree with Reviewer #3's opinion that our data could be interpreted as direct evidence for orientation/color facilitative computational mechanism at the dendritic level. We have clarified this point and modified the expression about functional trade-off in the revised manuscript.

3. They mapped the receptive field (RF) of dendritic inputs and found that the RF position varied across different dendritic input of same neuron. A previous cat area 17 study has also measured subthreshold RF inputs by in vivo intracellular recording, and they reported that size of subthreshold synaptic inputs (integration visual field) was about four times as large as the minimal discharge field (Bringuiet et al., Science 1999). Due to technical limitation, it would be impossible to map the RFs of all the input to individual, but this work gives an idea of subthreshold RF mapping on dendrites. Any possibility to get a large integration visual field based on the integration of local dendritic RFs over large cortical distance?

Re.: Thanks for this very insightful comment and suggestion. We have done this analysis and just as the reviewer predicted, we got quite a large integration field over the classical V1 neuronal receptive

fields (1.03 versus 0.34 Deg², Figure R12), thus providing a dendritic mechanism for the previously reported integration fields by Bringuier et al. (Science 1999).

Figure R12: Comparison of the local dendritic RF size versus the integration size. Each pair of dots represents data collected from one neuron.

4. Furthermore, this study compared the excitatory dendritic inputs on apical dendrites in upper layer with those on basal dendrites in the deeper layer. They found the apical dendritic inputs have similar but broader orientation preference with the basal dendritic inputs, and the apical dendritic inputs have larger RF size and lagged latency compared with basal dendritic inputs in the same neuron. It provides the first visualized evidence of temporal difference between dendrites in different depth. However, whether the recorded inputs on apical dendrites representing feedback signal and that on basal dendrite representing feedforward signal was not proved, at least not carefully discussed.

Re.: Yes, it remains challenging to directly prove input signals as feedback or feedforward ones, and our results could only bring strong but indirect evidence for this distinction. In future studies, we plan to obtain such direct evidence by combining new techniques, such as optogenetics or electrophysiology, and specific behavioral tasks, such as a top-down modulated selective attention task. We have discussed this point in greater detail in the revised manuscript.

5. This research provides a new, powerful and effective approach which can help understanding the

dendritic mechanism of signal processing on the surface of non-human primate cortex. Compared with the research of Garg et al., Science 2019 recording soma activities, this work provides much more details inside the neuronal dendrites and local circuits. However, the presentation of results (including the research background with specific scientific questions, the details of results, the description and discussion) is not clear enough. The manuscript needs major revisions. If it were presented as elegantly as Garg et al., Science 2019, the manuscript might be published already in Science.

Re.: Thanks for these constructive critiques and encouragements. We have tried our best to improve our manuscript in this revision.

Minor moments and suggestions:

1. The English writing is neither smooth nor flowing, because of the short format of the current version (it appears still in its previous Science format). Nature Communication will provide an idea platform for this work to be fully introduced, explained and discussed. The English itself must be clearer and need somebody with native English to improve it carefully and seriously.

Re.: We agree. We have taken this opportunity to present our work with a more fully developed introduction, description and discussion. In addition, native English speakers have helped us improve our grammar and the overall flow of ideas throughout the paper.

2. Which dots representing neuron I, II, III, and IV should be more clearly marked in Figure 2m, maybe also in Figure 2j and 2k.

Re.: Now clarified.

3. The figure legends should be presented more pithily. Some details of experiments could be moved into Method part, and some description of results could be moved into Result texture.

Re.: We agree. Now modified.

4. For the histogram plots in Figure 2 h & j, the scale of y axes should be 0~100, rather than 0~1 if it represents percentage (%).

Re.: Thanks, now corrected.

5. In Figure 4f (right), what is the meaning of red circle? It is not referred in the legend nor in the figure, does it represent the sampled neuron?

Re.: We apologize for the confusion. Yes, the red circle in the right panel of Fig. 4f designates the example neuron shown in the left panel of Fig. 4f. The revised figure legend now clarifies this.

6. The manuscript title could be changed to something like “From local to global: The integrated local dendritic inputs determine neural functional selectivity in primate V1”, which depicts what the authors have observed by applying their newly develop powerful technique to awake monkey V1.

Re.: We would be happy to implement this modification if the editor agrees.